# Foundation Models for Trajectory Planning in Autonomous Driving: A Review of Progress and Open Challenges

**Kemal Oksuz**                                                 *kemal.oksuz@bosch.com*
*Five AI Ltd., United Kingdom*
*Robert Bosch GmbH*

**Alexandru Buburuzan**                                  *alexandru.buburuzan@bosch.com*
*Five AI Ltd., United Kingdom*
*Robert Bosch GmbH*

**Anthony Knittel**                                          *anthony.knittel@bosch.com*
*Five AI Ltd., United Kingdom*
*Robert Bosch GmbH*

**Yuhan Yao**                                                    *yuhan.yao@de.bosch.com*
*Robert Bosch GmbH*

**Puneet K. Dokania**                                       *dokania.puneet@bosch.com*
*Five AI Ltd., United Kingdom*
*Robert Bosch GmbH*

**Reviewed on OpenReview:** *https://openreview.net/forum?id=E2L5J2O2Bk*

## Abstract

The emergence of multi-modal foundation models has markedly transformed the technology for autonomous driving, shifting away from conventional and mostly hand-crafted design choices towards unified, foundation-model-based approaches, capable of directly inferring motion trajectories from raw sensory inputs. This new class of methods can also incorporate natural language as an additional modality, with Vision-Language-Action (VLA) models serving as a representative example. In this review, we provide a comprehensive examination of such methods through a unifying taxonomy to critically evaluate their architectural design choices, methodological strengths, and their inherent capabilities and limitations. Our survey covers 37 recently proposed approaches that span the landscape of trajectory planning with foundation models. Furthermore, we assess these approaches with respect to the openness of their source code and datasets, offering valuable information to practitioners and researchers. We provide an accompanying webpage that catalogues the methods based on our taxonomy, available at: `https://github.com/fiveai/FMs-for-driving-trajectories`.

## Contents

---

The authors are from the ADAS Systems, Software & Services Business Unit of Robert Bosch GmbH.

# 1 Introduction

Foundation models (FMs) are large-scale models that leverage vast amounts of data to learn representations that can be effectively adapted to a variety of downstream tasks. Depending on the type of data they process, these models are generally referred to by different terms. FMs that operate only on language data, such as BERT (Devlin et al., 2019), GPT-2 (Radford et al., 2019), Chat-GPT (OpenAI, 2023) and Qwen (Bai et al., 2023a; Yang et al., 2024a; Qwen Team, 2025), are known as large language models (LLMs). Alternatively, models that process both language and visual data are considered vision language models (VLMs), with examples including CLIP (Radford et al., 2021), Flamingo (Alayrac et al., 2022), LLaVA (Liu et al., 2023; 2024a;b), GPT-4o (OpenAI, 2024), Intern-VL (Chen et al., 2024d; Gao et al., 2024c; Zhu et al., 2025) and Gemini (Gemini Team, 2025). As opposed to LLMs and VLMs, which generally output language data, a class of FMs are designed to generate images from language inputs (Saharia et al., 2022; Peebles & Xie, 2023), or from both language and vision inputs (Bruce et al., 2024; Genie Team, 2024; 2025; Meta Chameleon Team, 2025; NVIDIA, 2025).

These models generally serve as a backbone upon which more specialised models for various different domains are built using fine-tuning; autonomous driving (AD) is no exception. The scope of this work is to investigate the usefulness of FMs for AD. However, before delving into the AD-specific approaches, a natural question to ask would be: "*Can off-the-shelf FMs, specifically VLMs, understand driving scenarios without being explicitly trained or fine-tuned for them?*". To answer this, we prompt GPT-4o (OpenAI, 2024) with three complex driving scenes and related questions, and show the interaction in Fig. 1. The first two examples include rare driving scenes, where GPT-4o's replies are highly accurate and insightful. For example, in the first case, it not only understands the scene but also provides useful instructions on why and how one should drive cautiously in the given scenario. Similarly, in the last example, the model captures the misleading prompt and provides a reasonable explanation that promotes driving safely. Therefore, due to the scale of the architecture, training procedure, and the vast amount of training data involved, these models have learned representations that can already understand driving scenes surprisingly well, making them a highly promising precursor to build AD-specific solutions—a key driver to the paradigm shift currently observed in the AD industry. Nevertheless, to deploy FMs efficiently and reliably on edge devices for AD, several aspects of customising such models (e.g., chain-of-thought inference cost, scale, accessibility to open weights,

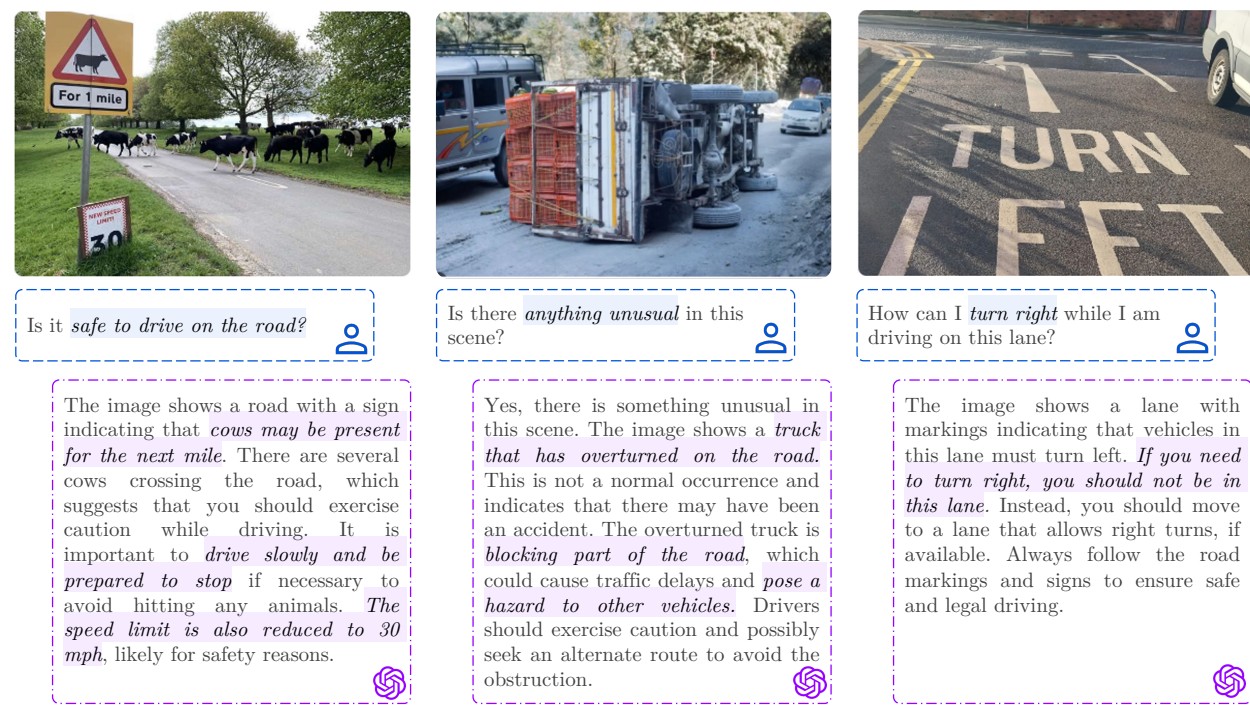

Figure 1: GPT-4o's response to driving-related prompts on three different scenarios.

training/inference efficiency, obtaining suitable data for fine-tuning, etc.) form integral parts of the ongoing research and design choices in this area. Through this review, we present a holistic perspective on these aspects, highlighting progress, limitations, and open research directions.

## 1.1 Scope and Contributions

FMs can be used in several ways for AD. Some methods leverage synthetic data for training (Chi et al., 2025; Yang et al., 2025a) and evaluation (Ljungbergh et al., 2024; Yan et al., 2025; Cao et al., 2025a), for which the models generating AD data, such as AD world models (Hu et al., 2023a; Wang et al., 2024b; Gao et al., 2024b; Wen et al., 2024b; Zhao et al., 2025a;c), can be helpful. A number of methods enhance scene understanding and reasoning capabilities of an FM for AD using visual question answering (VQA) tasks (Yang et al., 2024b; Ding et al., 2024b; Ma et al., 2024a; Nie et al., 2024; Lu et al., 2025; Qian et al., 2025a; Jiang et al., 2025c). Some methods also use an FM to improve a specific capability of the AD model such as perception (Pan et al., 2024b; Xinpeng et al., 2025), prediction (Zheng et al., 2024a; Zhou et al., 2025a) or control (Wang et al., 2023b; Sha et al., 2025). Within this context, a group of methods leverage FMs to yield textual actions, commonly by classifying the scene into one of the predefined meta-actions, such as *"go straight"* and *"slow down"* (Chen et al., 2023a; Fu et al., 2024; Wang et al., 2024c; Wen et al., 2024a; Jiang et al., 2025b; Ma et al., 2024b; Jin et al., 2024; Zhou et al., 2024b; Li et al., 2024a; Wang et al., 2025d). Last but not least, some models benefit from an FM to directly operate the vehicle, typically through trajectory planning (Pan et al., 2024a; Xu et al., 2025b; Tian et al., 2024; Fu et al., 2025a; Wang et al., 2025b; Hwang et al., 2025; Renz et al., 2025). While each of the above capabilities is essential for developing a robust and accurate AD model, trajectory planning is, arguably, the most critical task for driving, where the others serve as auxiliary functions for this primary goal. Consequently, considering the profound effect of FMs on this crucial task, in this paper, we primarily focus on how trajectory planning models for AD benefit from FMs.[1]

---

[1]Although we use the term trajectory planning, our scope also includes the models predicting outputs in different forms such as control signals. One example of this is DriveGPT4 (Xu et al., 2024), which predicts the target speed and turning angle of the autonomous vehicle.

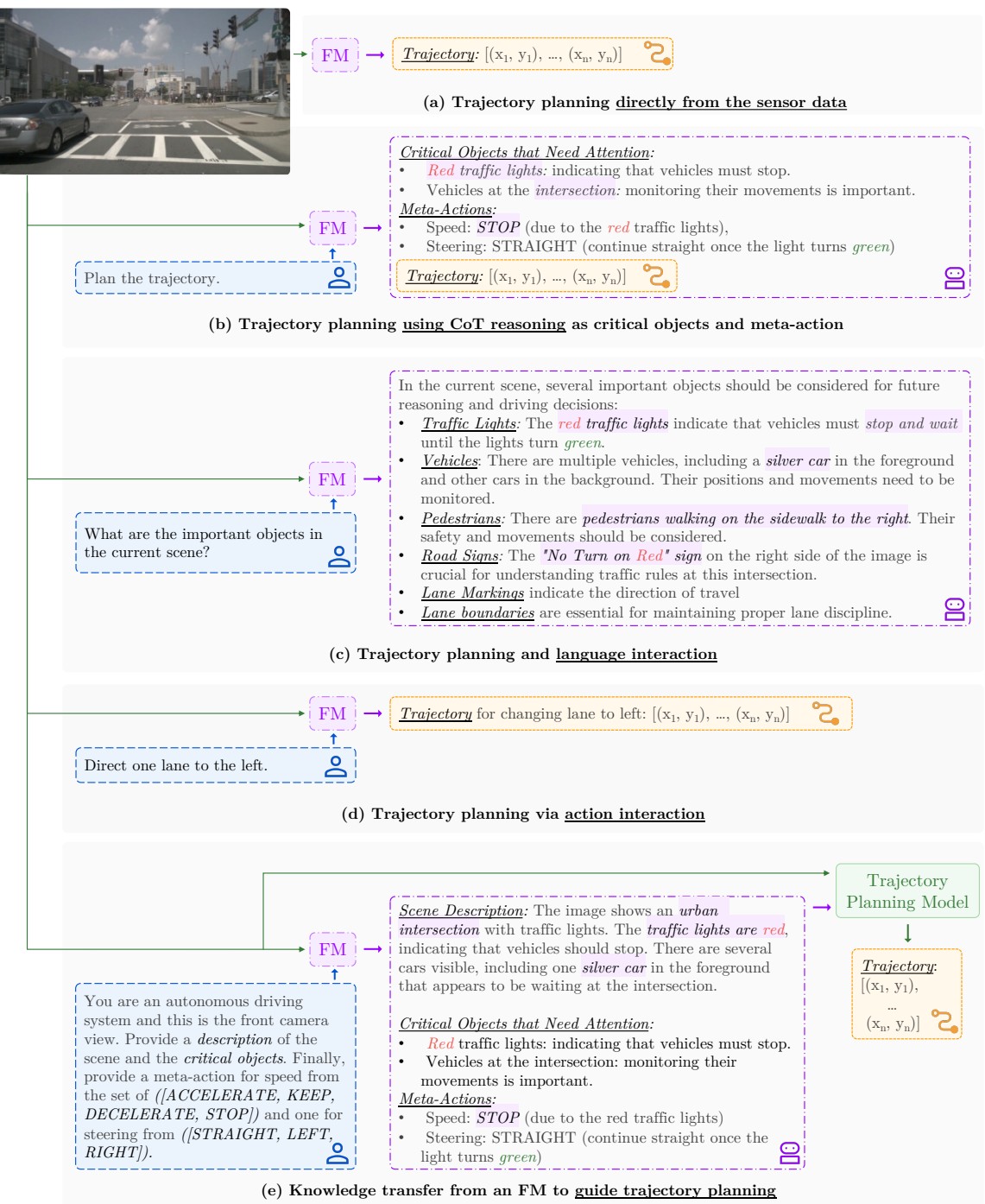

Figure 2: Different ways of how FMs are helping trajectory planning. While **(a-d)** provide examples of FMs tailored for trajectory planning, **(e)** is an example for an FM guiding trajectory planning. The input image is from nuScenes (Caesar et al., 2020), the question is from DriveLM-nuScenes (Sima et al., 2024), the user instruction is from the SimLingo dataset (Renz et al., 2025).

There are, in fact, various ways FMs can be leveraged for enhanced trajectory planning, as illustrated in Fig. 2. A common approach involves adapting existing FMs through minimal architectural modifications followed by fine-tuning them on a task-specific dataset. This fine-tuning can be as simple as training the model to output a trajectory directly from the sensor data (Mao et al., 2023; Zhang et al., 2024; Yuan et al.,

2024; Xu et al., 2025c; Xie et al., 2025; Zhou et al., 2025b)(see Fig. 2(a)). Additionally, some models use chain-of-thought (CoT) reasoning, a technique that helps LLMs break down a problem into multiple steps for enhanced reasoning. Fig. 2(b) illustrates its common usage, in which the FM first leverages CoT in the form of critical objects and meta-actions, before producing the final trajectory. The fact that VLMs are equipped with a linguistic modality has also motivated several approaches that leverage this capability to enable language and/or action interactions with trajectory planning models (Xu et al., 2024; Sima et al., 2024; Hwang et al., 2025; Shao et al., 2024a; Renz et al., 2025). As illustrated in Fig. 2(c), language interaction capability can offer reasoning behind a model's behaviour, giving reassurance to users. Differently, the action interaction capability facilitates driving assistance by executing driving commands of the user (refer to Fig. 2(d)), similar to existing vision language action model (VLA) models (Brohan et al., 2023; Kim et al., 2024). Alternatively, there are approaches where FMs *guide* an existing trajectory planning system by transferring their knowledge during training and/or inference (Pan et al., 2024a; Tian et al., 2024; Jiang et al., 2024; Wang et al., 2024a; Liu et al., 2025; Jiang et al., 2025a; Guo et al., 2025; Qian et al., 2025b; Xu et al., 2025b; Hegde et al., 2025; Chen et al., 2025; Han et al., 2025). Fig. 2(e) illustrates this set of approaches with an example, where a VLM (GPT-4o (OpenAI, 2024) in this case) outputs a description of the scene, critical objects that the ego should pay attention to, and a meta-action.

Although, as briefly indicated above, numerous studies benefit from FMs for trajectory planning in autonomous driving, the field still lacks a holistic understanding of the progress achieved. The sheer variety of existing heterogeneous approaches has made it increasingly difficult to tell what truly distinguishes one method from another, since the underlying architectures, datasets, and training procedures vary widely and have great impact on the performance. In this work, we aim to bring structure to this fragmented landscape by systematically analysing and comparing current techniques, highlighting the architectural design choices and capabilities that influence their effectiveness, and providing a unified perspective to guide further progress in applying foundation models to trajectory planning for autonomous driving. Our primary contributions, therefore, are:

1. We introduce a hierarchical taxonomy of the methods that employ FMs for trajectory planning in autonomous driving and systematically analyse 37 existing methods based on this taxonomy. *Our study is a comprehensive effort to organise, interpret, and unify this rapidly evolving field*, where a variety of heterogenous approaches has emerged without clear benchmarking and understanding of their distinctions. We believe that our study provides a structured foundation to help the field in making methodical progress.

2. In addition to *providing practical guidance* on how to tailor and to fine-tune an FM for trajectory planning and strategies for curating datasets for different use cases, *we also assess these approaches in terms of how open their code and data are*, in order to give practitioners and researchers useful pointers for their reproducibility and reuse. We also outline key future challenges and open research questions from multiple perspectives, including efficiency, robustness, evaluation benchmarks and sim-to-real transfer.

Given that pioneering works in this domain first emerged as preprints in late 2023 (e.g., GPT-Driver (Mao et al., 2023), DriveGPT4 (Xu et al., 2024)), this review encompasses approximately two years of research progress through October 2025 in leveraging FMs for autonomous trajectory planning. Within this period, we select 37 methods published in peer-reviewed venues to provide a clear conceptual analysis and well-structured organisation. We also include a limited number of representative preprints to incorporate emergent research. For example, V2X-VLM (You et al., 2024) represents a specialised FM for trajectory planning uniquely utilising additional infrastructure images, while FASIONAD++(Qian et al., 2025b) employs a FM selectively to guide trajectory planning only when the AD model has a high uncertainty. We emphasise that our objective is not to compare or rank these methods, but rather to organise and analyse them conceptually.

## 1.2 Comparison with Previous Reviews

Considering that there are several reviews or surveys focusing on AD, we would like to mention key differences between ours and the existing ones. One particular set of these reviews focuses on AD, usually to present

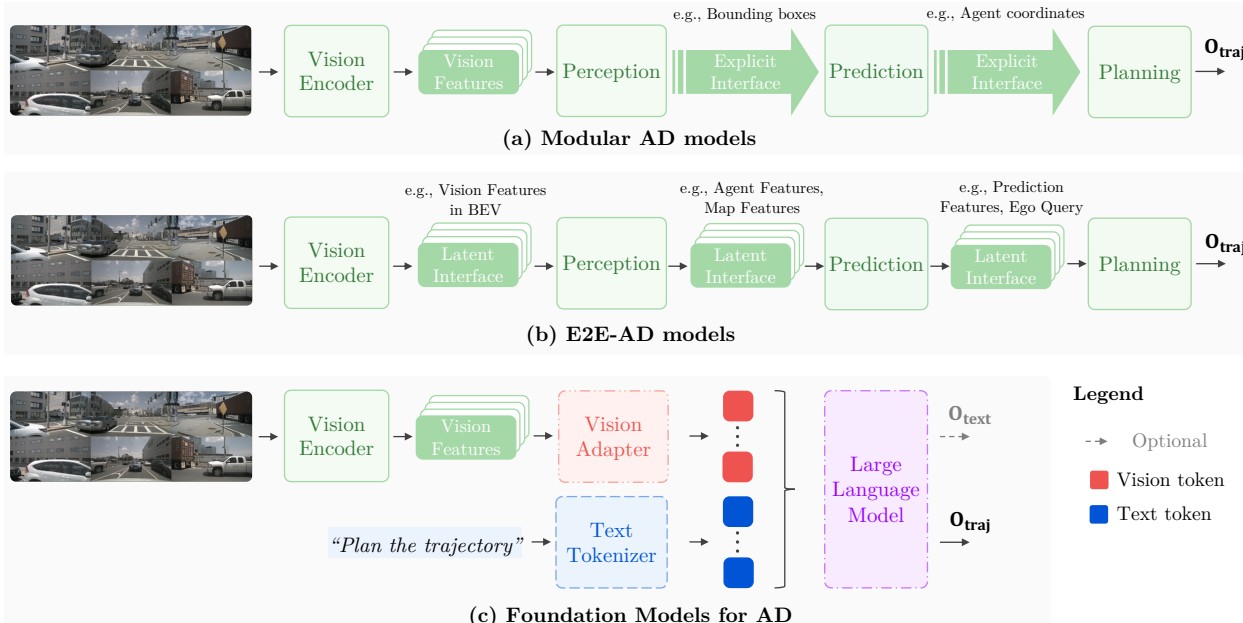

Figure 3: Trajectory planning methods. **(a) Modular** approaches use explicit interfaces between different modules. **(b) End-to-End** (E2E-AD) models replace explicit interfaces with latent ones, allowing all modules to be jointly differentiable. **(c) FM-based** methods that follow the typical VLM pipeline. The text output of the VLM can also be used, e.g., CoT reasoning. In this illustration, we simplify the pipelines to provide a high-level overview of how these models work, yet they can include different number of components and more complex connections across the modules. Input images are taken from nuScenes (Caesar et al., 2020).

and discuss the state-of-the-art at the time they were published (Yurtsever et al., 2020; Badue et al., 2021; Janai et al., 2021; Tampuu et al., 2022; Coelho & Oliveira, 2022; Li et al., 2023b; Zhao et al., 2024; Chen et al., 2024a). Some of these surveys have a narrower scope such as focusing on end-to-end (E2E)-trainable models (Tampuu et al., 2022; Coelho & Oliveira, 2022; Chen et al., 2024a), models using reinforcement learning (Kiran et al., 2022), imitation learning-based approaches (Le Mero et al., 2022), or existing AD datasets (Liu et al., 2024c). Alternatively, rather than trajectory planning, some reviews focus on a single auxiliary task for AD such as perception (Wang et al., 2025a), occupancy prediction (Xu et al., 2025a) or motion planning (Teng et al., 2023). While these are useful for their respective purposes, they do not pay special attention to how FMs can be used within the context of trajectory planning. As a result, our review is complementary to this set of reviews.

Due to the common adoption of FMs in several fields, including AD, a few review papers consider how these models can help AD (Gao et al., 2024a; Yang et al., 2024c;c; Zhou et al., 2024a; Li et al., 2025a; Cui et al., 2025b). Although they consider the use of FMs for AD, their scope is broader than ours. Specifically, these studies aim to cover how FMs can be useful from various perspectives, including perception, data generation, scene understanding, as well as trajectory planning. Consequently, the discussion on trajectory planning is quite limited, whereas we focus exclusively on trajectory planning, allowing a more comprehensive discussion. As one similar survey to ours, Jiang et al. (2025d) discuss different architectural paradigms for the models built on VLMs and their chronological progress. In addition, we include knowledge transfer approaches using FMs in our scope, introduce a hierarchical taxonomy over this broader set of models, delve deeper into the design choices for adapting a VLM for trajectory planning and elaborate on the openness of the methods that can be useful for researchers and practitioners.

## 2 Notations and Background

**Notations.** We denote the set of observations by $\mathbf{X}$, where $\mathbf{X}$ typically includes a subset of (i) sensor readings generally from cameras, lidar, or radar, (ii) the state of the ego vehicle such as its velocity, acceleration and steering, and (iii) an indicator of the route, which can be in the form of a GPS target point or a high level command such as {go straight, turn left, turn right}. We use $\mathbf{O_{traj}} \in \mathbb{R}^{k \times p}$ to represent the output trajectory, such that $k$ is the prediction horizon and $p$ is the dimension of the output at each time step, commonly $p = 2$ for bird's eye view (BEV) coordinates. Considering that the models typically aim to maximise the likelihood of the data, $\mathbf{O_{traj}}$ can be assumed to be a sample from the predictive distribution of the model, $\mathbf{O_{traj}} \sim p(\mathrm{O_{traj}}|\mathbf{X})$, where $\mathrm{O_{traj}}$ denotes the corresponding random variable. Furthermore, we use $\mathbf{O_{inittraj}}$ to denote an initially-predicted trajectory to account for methods predicting an initial trajectory before refining it to obtain $\mathbf{O_{traj}}$. $\mathbf{T}$ and $\mathbf{O_{text}}$ denote the input text to the FM and the output text from it, where each of them is a sequence of text tokens, i.e., $\mathbf{T} = \{\mathbf{T}^j\}_{j=1}^n$ and $\mathbf{O_{text}} = \{\mathbf{O}_{\mathbf{text}}^j\}_{j=1}^m$ with $n$ and $m$ being their sequence lengths.

FMs typically generate the next token of a text output conditioning on the previous tokens, known as the *autoregressive (AR) generation* or *next token prediction*. In the context of VLMs, which predict a categorical distribution over the text tokens, we formulate this as $\mathbf{O}_{\mathbf{text}}^i \sim p(\mathrm{O}_{\mathrm{text}}^i|\mathbf{X}, \mathbf{T}, \{\mathbf{O}_{\mathbf{text}}^j\}_{j=1}^{i-1})$ with $\mathrm{O}_{\mathrm{text}}^i$ being the random variable for the $i$-th token. For clarity, when we refer to the entire text output, we use $\mathbf{O_{text}} \sim p(\mathrm{O_{text}}|\mathbf{X}, \mathbf{T})$. Additionally, in the case of knowledge transfer (refer to Fig. 2(e)), the set of observations provided to the FM can be different from that provided to the trajectory planning model for AD. For example, while the AD model may receive multi-view images, the corresponding FM may receive only the front image, potentially with additional privileged information, e.g., VLM-AD (Xu et al., 2025b). As a result, while discussing the approaches with knowledge transfer, we denote the observations provided to the FM as $\mathbf{X_{FM}}$, and corresponding latent representations as $\mathbf{Z}$.

**Trajectory planning for AD.** The trajectory planning task for AD is typically considered as a non-deterministic environment where the world is partially observed through the sensors. Reinforcement Learning (Kazemkhani et al., 2025) and Tree-Search (Huang et al., 2024) methods typically formulate this environment as either a Fully- or Partially-Observed Markov Decision Process (Russell et al., 1995), and explicitly define numeric rewards and penalties to be optimised, while considering the distribution of expected rewards to form a control policy. In contrast, Imitation Learning delegates planning strategy to a teacher, where the objective of such a planner is to predict how a teacher model would act, and use this prediction as a plan for operating the vehicle. Essentially, given a set of observations $\mathbf{X}$ from the sensors, the trajectory planning task involves determining a trajectory for the ego vehicle to follow ($\mathbf{O_{traj}}$). And, to operate a vehicle, $\mathbf{O_{traj}}$ is typically propagated to a controller such as a Proportional-Integral-Derivative (PID) controller (Bennett, 2001) to obtain control signals, such as for the accelerator and steering.

Model architectures for trajectory planning can be divided into three main groups, as shown in Fig. 3. Modular approaches, shown in Fig. 3(a), use explicit interfaces, where each module independently aims to solve a subproblem such as perception (Zhu et al., 2021; Zeng et al., 2022; Liu et al., 2022; Wang et al., 2023a; Oksuz et al., 2023; Yavuz et al., 2024; Li et al., 2025b), prediction (Shi et al., 2022; 2023; Zhou et al., 2023b; Cheng et al., 2023; Prutsch et al., 2024) or planning (Gao et al., 2020; Chen et al., 2024b). This architecture does not allow gradient flow during training, i.e., it is not E2E-trainable, and each module is optimised for its own objective. E2E-AD models in Fig. 3(b) replace explicit interfaces by latent representations from the previous module, to optimise all modules jointly for trajectory planning (Hu et al., 2023b; Jiang et al., 2023; 2026; Li et al., 2024c;b; Weng et al., 2024; Zheng et al., 2024b; Liao et al., 2025; Zhang et al., 2025; Song et al., 2025). FM-based models are recent approaches that are built directly on FMs to benefit from their vast world knowledge (Xu et al., 2024; Zhang et al., 2024; Wang et al., 2025b; Renz et al., 2025; Hwang et al., 2025; Fu et al., 2025a; Xie et al., 2025; Chen et al., 2025). Fig. 3(c) illustrates a typical VLM pipeline (Bai et al., 2023b; Li et al., 2023a; Liu et al., 2023; 2024b), in which a vision encoder (He et al., 2016; Dosovitskiy et al., 2021; Liu et al., 2021) outputs vision features, which are then projected onto the LLM embedding space using a vision adapter, e.g., a linear layer (Liu et al., 2023). Finally, an LLM processes vision and text tokens to yield the final trajectory. As more recent approaches, existing E2E-AD and FM-based methods are predominantly trained using Imitation Learning.

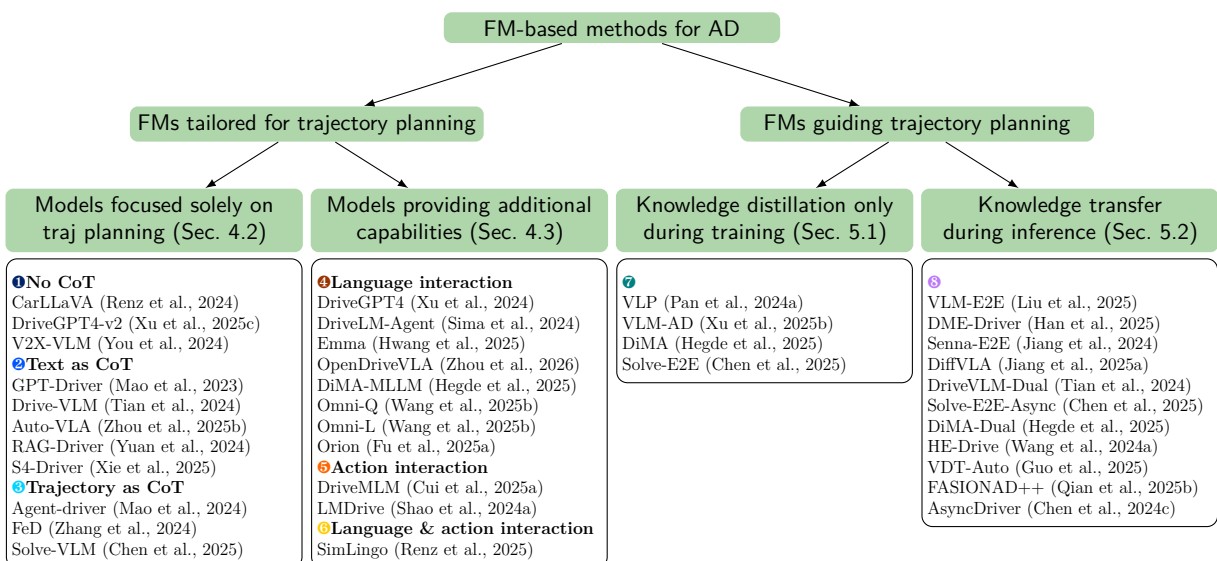

Figure 4: Taxonomy of trajectory planning methods utilising or getting help from FMs.

# 3   A Hierarchical Taxonomy

We now present a taxonomy to systematically study trajectory planning approaches utilising FMs. Broadly speaking, there are two main categories: FMs *tailored for* trajectory planning, and FMs *guiding* trajectory planning. However, these main categories do involve various subcategories, eventually forming a hierarchical taxonomy as shown in Fig. 4. Specifically, each of the eight leaf nodes in the taxonomy tree corresponds to the formulations in Fig. 5. In what follows, we discuss them in detail.

## 3.1   Foundation Models Tailored for Trajectory Planning

The primary characteristic of the methods in this group is that they, partly or fully, utilise existing pre-trained FMs by tailoring and fine-tuning them to trajectory planning for AD. Therefore, effectively, they build FMs that are directly used for AD use cases. Considering the auto-regressive generation of FMs and the convention that the trajectory $\mathbf{O_{traj}}$ is commonly predicted after the text output $\mathbf{O_{text}}$, these approaches can be formulated as:

$$\mathbf{O_{text}} \sim p(O_{text}|\mathbf{X}, \mathbf{T}) = f(\mathbf{X}, \mathbf{T}), \text{ and } \mathbf{O_{traj}} \sim p(O_{traj}|\mathbf{X}, \mathbf{T}, \mathbf{O_{text}}) = f(\mathbf{X}, \mathbf{T}, \mathbf{O_{text}}), \quad (1)$$

where $f(\cdot)$ is the fine-tuned FM. By design, these methods have the advantage of directly exploiting the vast world knowledge of FMs and tailor it towards autonomous driving applications via fine-tuning, a significant advantage over traditional methods for trajectory planning. Additionally, the choice of FM also allows these models to have new capabilities. For example, in the case of language-based FMs, capabilities such as language- and action-based interactions with users are possible—along with the original task of trajectory planning—during inference. Further details regarding the design choices and categorisation of these methods are provided in Sec. 4.

We identify 22 current approaches from the literature falling under this broader category. Below, we divide them further into two subgroups depending on the features they have.

**Models focused solely on trajectory prediction**. These models either have no text prompt $\mathbf{T} = \emptyset$, or a fixed one. In the latter case, $\mathbf{T}$ simply aims to trigger the model to yield the desired output, e.g., the fixed prompt could be "*plan the trajectory*". Since there is no variability in the language, $\mathbf{O_{traj}}$ primarily relies on input observations $\mathbf{X}$. Methods using CoT reasoning solely to provide better trajectories during inference also fall in this category. Similarly, these methods either use a *fixed* single prompt or a set of multiple sequential *fixed* prompts in a pre-defined order to exploit reasoning abilities of FMs. However, the choice of the nature of

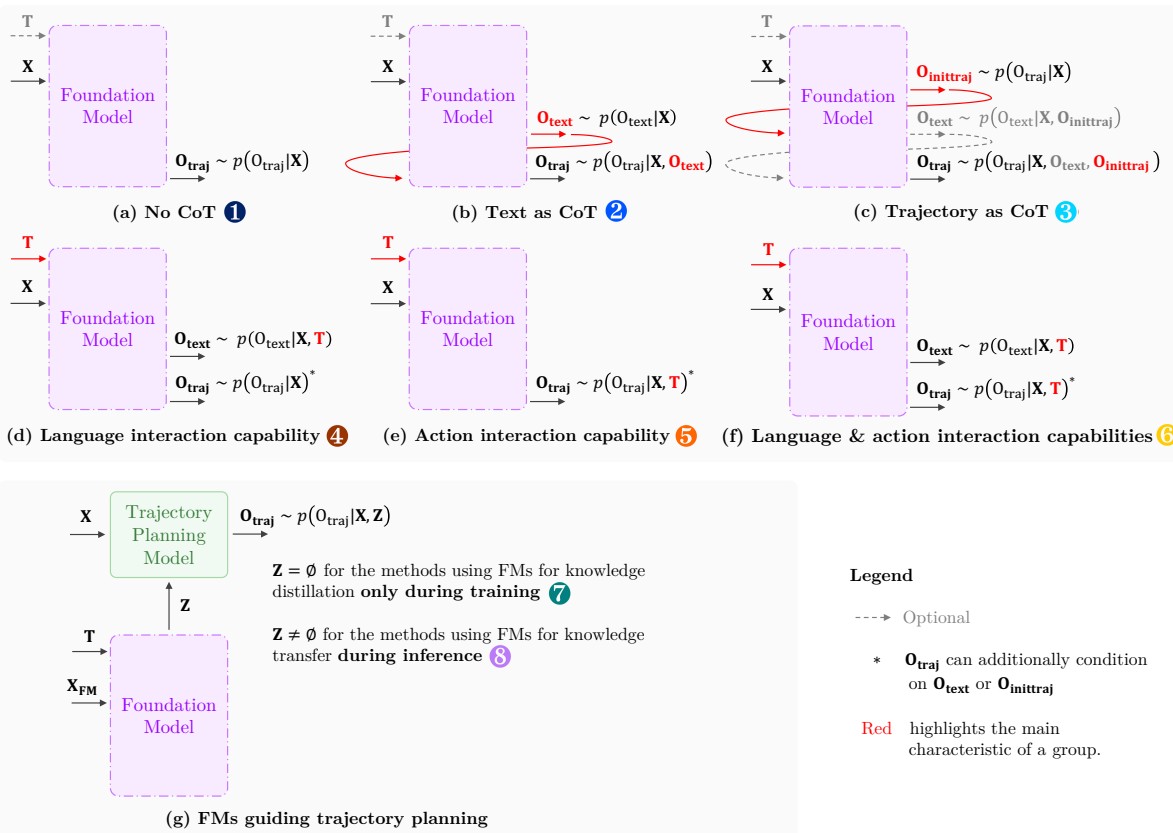

Figure 5: Formulations of subcategories in our taxonomy. **(a–f)** FMs *tailored* for trajectory planning. Specifically, **(a–c)** are focused solely on trajectory planning. These methods either do not have a text prompt **T** or have a fixed one, hence **T** is optional. (c) shows a case where $\mathbf{O_{traj}}$ optionally conditions on the text output $\mathbf{O_{text}}$. Similarly, $\mathbf{O_{inittraj}}$ can also benefit from a text output as CoT, which is omitted for clarity. **(d–f)** are the subcategories providing additional capabilities. In (d–f), $\mathbf{O_{traj}}$ is shown without CoT for clarity and $^*$ highlights that $\mathbf{O_{traj}}$ can also be obtained using different forms of CoT in (b) or (c). **(g)** FMs *guiding* trajectory planning via knowledge distillation.

these prompts plays a crucial role in understanding different factors influencing the decision making of these models, therefore, we further divide this category into three subcategories depending on how, if, CoT is used.

- ❶ No CoT, hence $\mathbf{O_{text}} = \emptyset$ and $\mathbf{O_{traj}} \sim p(\mathrm{O_{traj}}|\mathbf{X})$ (see Fig. 5(a)). Fig. 2(a) sets an example for this category. As an example, given front view image, speed of the ego and GPS target points, CarLLaVA (Renz et al., 2024) only outputs the trajectory.
- ❷ Text outputs for CoT, hence $\mathbf{O_{text}} \sim p(\mathrm{O_{text}}|\mathbf{X})$ and $\mathbf{O_{traj}} \sim p(\mathrm{O_{traj}}|\mathbf{X}, \mathbf{O_{text}})$ (see Fig. 5(b)). Fig. 2(b) can be considered as an example of this category. One example of this subcategory is DriveVLM (Tian et al., 2024), where a text output including a scene description (e.g., weather, road type, critical objects), influence of the critical objects on ego and planning decision are generated before the model outputs the trajectory.
- ❸ Initial trajectory prediction $\mathbf{O_{inittraj}}$ for CoT, hence predicting $\mathbf{O_{traj}} \sim p(\mathrm{O_{traj}}|\mathbf{X}, \mathbf{O_{text}}, \mathbf{O_{inittraj}})$ (refer Fig. 5(c)). Such models first yield $\mathbf{O_{inittraj}}$ and then refine it further to $\mathbf{O_{traj}}$. These models can still leverage text output as well for CoT reasoning while generating either $\mathbf{O_{inittraj}}$ or $\mathbf{O_{traj}}$. To illustrate, FeD (Zhang et al., 2024) aims to improve the initially-predicted trajectory based on the textual feedback it generates, e.g., if $\mathbf{O_{inittraj}}$ could result in a collision.

**Models providing additional capabilities.** In addition to trajectory planning, these models exploit the language understanding of FMs to build additional features to cater for language- and/or action-based interactions. Below we further divide them into three subcategories.

- ❹ Language interaction capabilities via $\mathbf{O_{text}} \sim p(\mathrm{O_{text}}|\mathbf{X}, \mathbf{T})$ (see Fig. 5(d)). These language interaction capabilities primarily involve question-answer or description tasks, generally for the purpose of informing users about the surrounding or the potential action taken by the model, as also illustrated in Fig. 2(c). These capabilities also serve as a tool to understand the decision-making process of these models—allowing practitioners to provide explanations towards model's actions, debug model's outcomes, or to potentially reassure users about its behaviour. As an example, Orion (Fu et al., 2025a) can respond to different questions such as "Could you describe the overall environment and objects captured in the images provided?" and "Has the traffic light influenced the driving strategy of the ego vehicle in the previous frames?". Accordingly, the set of the text inputs $\mathbf{T}$ that such models can respond is richer than the models providing text output as CoT, in which $\mathbf{T}$ typically does not exist or it can be limited to a single prompt, e.g., *"What is my future trajectory?"* (Xie et al., 2025). The trajectory planning $\mathbf{O_{traj}}$ in these models can follow one of the alternatives in Fig. 5(a-c) depending on if CoT reasoning is used.
- ❺ Action interaction capabilities where text prompt $\mathbf{T}$ may directly result in modifications to the planned trajectory by the model (see Fig. 5(e)). For example, an input that prompts a model to **follow instructions** would be *"turn left from the next intersection"* and this prompt may result in replanning of the predicted trajectory. Such instruction following prompts indicate the model to behave in a particular way and are normally diverse in nature. Few examples from LMDrive (Shao et al., 2024a) include *"Feel free to start driving.", "Depart at the second exit on the roundabout.",* and *"Execute a right maneuver, prepare for highway exit"*. These are much more diverse compared to the navigational instructions (or route indicators) which consist of pre-defined set of basic commands such as *turn right* or *go straight*, therefore, we do not consider models that can only interact with a navigator in this category. Note, in addition to instructions that directly impact actions, other forms of instructions can surely have indirect impact. For example, an input *"watch out for crossing pedestrians"* that provides a warning would be considered as a **notice instruction** prompt, while *"crash into the vehicle front"* would be considered as a **misleading instruction** prompt where the model is expected to avoid following this prompt and follow a safe trajectory.
- ❻ Some models can also have both action and language interaction capabilities, as illustrated in Fig. 5(f). SimLingo (Renz et al., 2025) is the only example falling under this subcategory among the methods we consider in this paper.

## 3.2 Foundation Models Guiding Trajectory Planning

Inspired by the well-known knowledge distillation work (Hinton et al., 2014), these models do not build an FM for driving use cases, rather they mostly use off-the-shelf FMs to help improve their existing trajectory planning models for AD. From a broader perspective, the methods falling under this category can be formulated as,

$$\mathbf{O_{traj}} \sim p(\mathrm{O_{traj}}|\mathbf{X}, \mathbf{Z}) = \mathrm{f}(\mathbf{X}, \mathbf{Z}), \tag{2}$$

where $\mathbf{Z}$ is the transferred knowledge from the FM to the corresponding AD model $\mathrm{f}(\cdot, \cdot)$. Note, in this case, $\mathrm{f}(\cdot, \cdot)$ is either a modular approach (Chen et al., 2024c) (see Fig. 3(a)), where the FM can be used at any level to improve trajectory planning, or an E2E-AD model (see Fig. 3(b)) (Hu et al., 2023b; Jiang et al., 2023).

We identify 15 methods in this category and further split them into two subcategories depending on whether $\mathbf{Z}$ is needed during inference or not, as illustrated in Fig. 5(g).

❼ **Knowledge distillation only during training.** These approaches employ an FM for knowledge distillation during training the AD model. As an example, this can be achieved by prompting a VLM with a text input and sensor data to obtain a structured output such as a meta-action, which can be distilled into the AD model by appending a meta-action prediction module to it, similar to VLM-AD (Xu et al., 2025b). In such a case the FM is not needed for inference, effectively corresponding to $\mathbf{Z} = \emptyset$ in Eq. (2). Hence, the prediction of the model is conditioned only on the observations $\mathbf{X}$ during inference, i.e., $\mathbf{O_{traj}} \sim p(\mathrm{O_{traj}}|\mathbf{X})$.

This offers the advantage of maintaining the inference efficiency of the AD model as the FM is not needed for inference.

**8** **Knowledge transfer during inference.** These methods utilise an FM not only during training, but also during inference with the intention to leverage the knowledge of FMs more effectively. This corresponds to $\mathbf{Z} \neq \emptyset$ in Eq. (2), where $\mathbf{Z}$ is usually taken as *a scene description* (Liu et al., 2025), typically involving perception knowledge such as the objects in the scene, or *a planning decision*, which can include a trajectory (Tian et al., 2024; Chen et al., 2025) or a meta-action (Jiang et al., 2024) from the FM. In either of the cases, $\mathbf{Z}$ is typically used either as an internal representation of the FM, or directly as the output of the FM. In the latter, the FM output can also be a plain text, in which case an additional text encoder is typically employed for encoding the text before propagating to the AD model. Nevertheless, due to the use of FM at inference, this group of approaches typically requires additional compute for inference compared to the methods mentioned in the category above.

# 4 Foundation Models Tailored for Trajectory Planning

We now delve deeper into the design and development of FMs tailored for trajectory planning for AD. We first elaborate on the important ingredients one must pay attention to before even beginning to fine-tune an FM for trajectory planning in Sec. 4.1, and then discuss different approaches for each of the subcategories in detail in Sec. 4.2 and Sec. 4.3.

## 4.1 How to Fine-tune an FM for Trajectory Planning

**Data Curation.** The structure of the data to be curated while adapting an FM for trajectory planning depends primarily on the desired use case of the model, as illustrated in Fig. 6(a).

- *Driving Dataset.* The dataset for trajectory planning would simply comprise the pairs of observations and the trajectory, i.e., $(\mathbf{X}, \mathbf{O_{traj}})$. As aforementioned, the observations $(\mathbf{X})$ typically include a subset of the sensor data, the ego status, and an indicator of the route. An example of this form of dataset is nuScenes (Caesar et al., 2020), including multi-view camera, radar and lidar data as sensor input, and future ego positions as the trajectory target.
- *Driving Dataset with CoT Reasoning.* For complex situations where reasoning is crucial, one might be interested in exploiting the CoT reasoning ability of FMs for enhanced understanding of the scene. To enable that, each tuple in the driving dataset is extended by $\mathbf{O_{text}}$ (the text to enforce CoT reasoning) leading to the dataset consisting of tuples $(\mathbf{X}, \mathbf{O_{text}}, \mathbf{O_{traj}})$. An example dataset with textual descriptions that can be used as CoT reasoning is BDD-X (Kim et al., 2018). When CoT is performed using an initial trajectory $\mathbf{O_{inittraj}}$, a dataset might not be curated and stored in advance, and instead it is generated by the model. These driving datasets could also include a typically fixed input prompt $\mathbf{T}$, such as the example in Fig. 2(b) with $\mathbf{T} =$ *"Plan the trajectory"*.
- *Language Interaction Capability.* Although this resulting driving dataset can be used for trajectory planning, additional tuples are needed to account for the language or action interaction capabilities, depending on the use case. Specifically, for the language interaction capability, the dataset is extended with new tuples, $(\mathbf{X}, \mathbf{T}, \mathbf{O_{text}})$ where $\mathbf{T}, \mathbf{O_{text}}$ is the question-answer pair grounding on $\mathbf{X}$. An example dataset with language interaction is the Chat-B2D dataset used by Orion (Fu et al., 2025a).
- *Action Interaction Capability.* If the model is expected to generate a trajectory, while answering the question, then such tuples include $\mathbf{O_{traj}}$, i.e., $(\mathbf{X}, \mathbf{T}, \mathbf{O_{text}}, \mathbf{O_{traj}})$, in which case the answer $\mathbf{O_{text}}$ could serve as CoT reasoning. As for the action interaction capability, the model is expected to consider the user instruction for trajectory planning. Accordingly, the data tuples typically follow $(\mathbf{X}, \mathbf{T}, \mathbf{O_{traj}})$, such that $\mathbf{T}$ is the user instruction and $\mathbf{O_{traj}}$ is the trajectory corresponding to $\mathbf{T}$. An example dataset with language and action interaction is the SimLingo dataset (Renz et al., 2025). Again, optionally, $\mathbf{O_{text}}$ can be included for CoT reasoning during instruction following. We further discuss the different choices of the CoT reasoning in Sec. 4.2 and the datasets designed to include additional capabilities in Sec. 4.3.

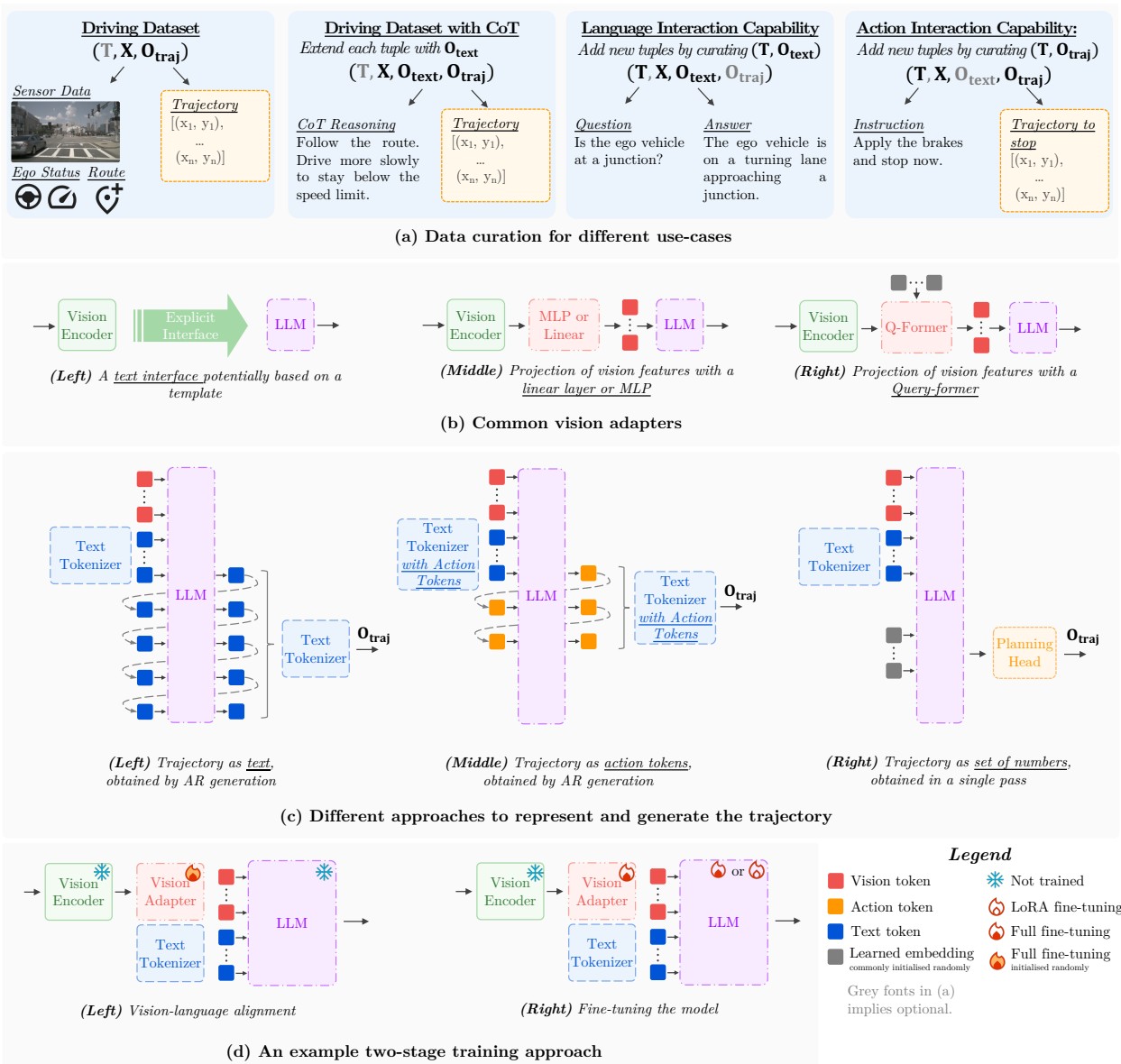

Figure 6: The steps to fine-tune an FM for trajectory planning. **(a) Data curation. (b)** For the **model design**, either an off-the-shelf VLM can be used or a vision encoder can be combined with an LLM using one of the depicted vision adapters. **(c) Trajectory representation.** Vision encoder and adapter are omitted for clarity. **(d)** Among various **model training strategies**, this is an example two-stage training approach based on Liu et al. (2023), usually adopted when an off-the-shelf VLM is **not** used.

**Model Design.** While designing the model architecture, some approaches (Renz et al., 2025; Zhou et al., 2025b) use off-the-shelf VLMs, providing the advantage of initialising all parameters jointly from a state trained on a large dataset. Differently, another group of approaches (Xu et al., 2024; Chen et al., 2025; Fu et al., 2025a) combines a preferred vision encoder, such as ViT (Dosovitskiy et al., 2021), with a preferred LLM using a randomly-initialised vision adapter. Both approaches follow the typical VLM architecture (see Fig. 3(c)), where the vision adapter connects the vision encoder and the LLM. Additionally, the latter approach to combine preferred vision encoder and LLM, requires designing a vision adapter. As shown in Fig. 6(b), in general, there are three different choices of the vision adapter:

- The vision adapter can be a *text interface* where the vision encoder directly outputs perception and/or prediction information such as the locations of the objects and map elements in the scene. This information is then propagated to the LLM as text, in which case the model is not trained end-to-end as in Mao et al. (2023).
- *A linear layer or an multi-layer perceptron (MLP)* can project all vision tokens from the vision encoder onto the LLM input space (Liu et al., 2023) as used by Renz et al. (2025).
- Different from a linear layer or MLP block, *Queryformer* (Li et al., 2023a) relies on cross-attention between a set of typically randomly-initialised latent representations and the tokens from the vision encoder to selectively project the most useful set of representations. This approach is taken by Omni-Q (Wang et al., 2025b).

Additionally, including custom modules can impose certain inductive biases absent in off-the-shelf FMs. For example, as off-the-shelf VLMs are generally not pretrained on 3D perception tasks or using videos, some approaches design custom modules (Fu et al., 2025a; Wang et al., 2025b; Xie et al., 2025; Chen et al., 2025) to enforce the model to consider these clues absent in the VLMs.

**Trajectory Representation.** Another crucial aspect in tailoring an FM for AD is how to represent the trajectory, as this can consequently require modifications in the model design as well. Fig. 6(c) illustrates common trajectory representations, which we elaborate on below:

- *Trajectory as text, obtained by the standard AR text generation of the LLM.* For example, consider a model that generates a single character in each pass and assume that $1.54, 0.21$ is a 2D point of the trajectory. As a result, this 2D point is generated sequentially as '1', '.', '5', '4', ',', '0', '.', '2', '1' in an AR manner, as illustrated in Fig. 6(c)(Left). Consequently, while this approach generally does not require any modification to the tokenizer or model architecture, it requires multiple passes through the LLM to yield a single point of the trajectory, increasing the inference time.
- *Trajectory as action tokens.* Instead of generating a single 2D point in multiple passes, one can discretise the action space and represent these discrete actions in the vocabulary of the LLM by the action tokens. One example is to discretise 2D BEV space using a grid and use the points on the grid as the action tokens (Sima et al., 2024; Brohan et al., 2023). These action tokens can be included in the tokenizer by mapping the rarely-used tokens in the vocabulary to each of these 2D points (Brohan et al., 2023). Zhou et al. (2025b) also follow a similar approach by building a vocabulary for the actions with 2048 actions determined by clustering the actions in terms of the relative 2D position and the heading angle in the next 0.5 seconds. In these examples, the trajectory consists of multiple points, and hence multiple AR passes within the LLM are required to obtain a trajectory, as shown in Fig. 6(c)(Middle). As an alternative, similar to the planning vocabulary in VADv2 (Jiang et al., 2026), an action token can represent an entire trajectory where the trajectory can be obtained in a single pass, however, this might be more prone to errors due to long-horizon planning.
- *Trajectory as a set of numbers, generally obtained in a single pass by using a planning head.* Alternatively, the trajectory can be obtained by an additional planning head that utilises the representations learned by the LLM. There are two main variants of this approach. As shown in Fig. 6(c)(Right), some methods (Renz et al., 2024; Zhang et al., 2024; Renz et al., 2025) use learned embeddings that attend to the observations (and a text input, if exists) before being decoded as a trajectory by the planning head. On the other hand, another group of approaches (Fu et al., 2025a; Xu et al., 2025c) does not introduce learned embeddings, and instead the planning head relies on the representations of the LLM in the last layer. In any case, this approach requires designing a planning head to be appended to the LLM. MLP (Renz et al., 2025) and generative models, such as a variational autoencoder (Fu et al., 2025a) or diffusion model (Liao et al., 2025), have been explored in the literature for this purpose.

**Model Training.** Fine-tuning the FM is typically carried out in a single step or in multiple steps, where each step generally targets a specific subset of the modules. For example, similar to LLaVA-style training (Liu et al., 2023), as illustrated in Fig. 6(d), one can first train the vision adapter by freezing the vision encoder and LLM, especially when a custom vision adapter is introduced with randomly-initialised parameters. After training the adapter, all model parameters or a subset of them can be fine-tuned for the target task. To provide an overview of how the existing methods in the literature fine-tune their models, we introduce the

Table 1: Design choices of FMs tailored for AD (refer to Fig. 4). Unless mentioned otherwise in the corresponding paper, we assume that each method uses full parameter fine-tuning of a module, generates the trajectory in the form of text, and does not have route indicator as an input. Custom VLM implies that the model does not employ an off-the-shelf VLM but combines a vision encoder with an LLM itself, in which case we specify the details of the vision encoder and LLM, e.g., SigLIP-ViT-G as in the case of S4-Driver. Please refer to Sec. 4 for the details of the training symbols.

| | | Observations | | Model Design and Model Training | | | | | |
|---|---|---|---|---|---|---|---|---|---|
| | Model | Sensors | Route indicator | VLM | Vision Encoder | Vision Adapter | LLM | # Params | Trajectory Representation |
| 1 | CarLLaVA (Renz et al., 2024) | Front camera | ✓ | LLaVA-Next | ViT-L 💧 | Linear 💧 | Tiny-Llama 💧 | >350M | Numbers |
| ❶ 2 | DriveGPT4-v2 (Xu et al., 2025c) | 3 Front cameras | ✓ | Custom | SigLIP-ViT-L ❄️ | Linear 🔥 | Qwen-0.5B 💧 | >1B | Numbers |
| 3 | V2X-VLM (You et al., 2024) | Front+Infra camera | ✗ | Florence-2 | DaViT ❄️ | Linear 🔥 | BART 🔥 | 232B | Text |
| 4 | GPT-Driver (Mao et al., 2023) | 360° cameras | ✓ | Custom | Based on UniAD ❄️ | Text | Chat-GPT3.5 💧 | >175B | Text |
| 5 | Drive-VLM (Tian et al., 2024) | Front camera | ✗ | Qwen-VL | Custom ViT 🔥 | MLP 🔥 | QwenLM 🔥 | 9.6B | Text |
| ❷ 6 | Auto-VLA (Zhou et al., 2025b) | 3 Front cameras | ✓ | Qwen2.5-VL | Custom ViT 🔥 | MLP 🔥 | Qwen2.5 🔥 | >3B | Action token |
| 7 | RAG-Driver (Yuan et al., 2024) | Front camera | ✗ | Video-LLaVA | ViT-B ❄️ | Linear 🔥 | LLaMA2 🔥 | >7B | Text |
| 8 | S4-Driver (Xie et al., 2025) | 360° cameras | ✓ | PaLI-3 | SigLIP-ViT-G ❄️ +Custom 🔥 | Linear 🔥 | UL2 🔥 | >5B | Text |
| 9 | Agent-Driver (Mao et al., 2024) | 360° cameras | ✓ | Custom | Based on UniAD ❄️ | Text | Chat-GPT3.5 💧 | >175B × 2 | Text |
| ❸ 10 | FeD (Zhang et al., 2024) | Front camera | ✓ | LLaVA | ViT-L 💧 | Linear 💧 | Llama 💧 | >7B | Numbers |
| 11 | Solve-VLM (Chen et al., 2025) | 360° cameras | ✗ | Custom | EVA-02-L 🔥 | Q-Former 🔥 | LLaVA-v1.5-LLM 💧 | >7B | Text |
| 12 | DriveGPT4 (Xu et al., 2024) | Front camera | ✗ | Custom | Valley-ViT-L ❄️ | Valley-Custom 🔥 | Llama2 🔥 | >7B | Text |
| 13 | DriveLM-Agent (Sima et al., 2024) | Front camera | ✗ | BLIP-2 | ViT-L 💧 | Q-Former 💧 | Flan-T5 💧 | >7B | Action token |
| 14 | Emma (Hwang et al., 2025) | 360° cameras | ✓ | Gemini | Unknown 💧 | Unknown 💧 | Unknown 💧 | 1.8B | Text |
| ❹ 15 | OpenDriveVLA (Zhou et al., 2026) | 360° cameras | ✓ | Custom | Bevformer 🔥 | Q-Former+MLP 🔥 | Qwen 2.5-Instruct 💧 | >0.5B-7B | AR text |
| 16 | DiMA-MLLM (Hegde et al., 2025) | 360° cameras | ✗ | Custom | Bevformer 🔥 | Q-Former 🔥 | LLaVA-v1.5-LLM 💧 | >7B | Text |
| 17 | Omni-L (Wang et al., 2025b) | 360° cameras | ✓ | Custom | EVA-02-L 🔥 | MLP 🔥 | LLaVA-1.5-LLM 💧 | >7B | Text |
| 18 | Omni-Q (Wang et al., 2025b) | 360° cameras | ✓ | Custom | EVA-02-L 🔥 | Q-Former 🔥 | LLaVA-1.5-LLM 💧 | >7B | Text |
| 19 | Orion (Fu et al., 2025a) | 360° cameras | ✓ | Custom | EVA-02-L 🔥 | Q-Former 🔥 | Vicuna-1.5 💧 | >7B | Numbers |
| ❺ 20 | DriveMLM (Cui et al., 2025a) | 360° cameras, lidar | ✓ | Custom | EVA-ViT-G ❄️ +GD-MAE ❄️ | Q-Former 🔥 | Llama 🔥 | >7B | Text |
| 21 | LMDrive (Shao et al., 2024a) | 360° cameras, lidar | ✓ | Custom | ResNet ❄️ +PointPillars ❄️ | Q-Former 🔥 | LLaVA-v1.5-LLM ❄️ | >7B | Numbers |
| 22 | SimLingo (Renz et al., 2025) | Front camera | ✓ | Mini-InternVL | InternViT 💧 | MLP 💧 | Qwen2 💧 | 1B | Numbers |

following symbols that represent the change of the parameters in a module across training (spanning all stages in the case of multi-stage training):

- ❄️ indicates keeping modules completely frozen during training.
- 💧 represents low-rank adaptation (LoRA)-style fine-tuning of a module (Hu et al., 2022), which assumes that the adaptation needed for new tasks can be captured by low-rank updates. This approach is efficient but limits the capacity of the model for adaptation.
- 🔥 implies the standard fine-tuning of all parameters, utilising the full capacity of the module for fine-tuning, though it requires more resources than using LoRA.
- 🔶 represents training a module by initialising it either randomly, e.g., once a custom module is introduced, or from a model pretrained only on domain-specific data, e.g., a vision encoder trained only on nuScenes (Caesar et al., 2020).

Using these symbols, Tab. 1 presents the main design choices of 22 methods within the scope of this section.

## 4.2 Models Focused Solely on Trajectory Planning

Following the taxonomy provided in Fig. 4, we now discuss different subcategories. Tab. 2 provides an example approach for each of these subcategories.

Table 2: Example models solely focused on trajectory planning. For such models, $\mathbf{T}$ is either $\emptyset$ or fixed, and CoT reasoning includes $\mathbf{O_{text}}$ or an initially-planned trajectory $\mathbf{O_{inittraj}}$ for self-correction.

| Model Type | Example Model | Observations (X) | System Prompt (T) | Text Output (O_text) |
|---|---|---|---|---|
| $\mathbf{O_{traj}} \sim p(\mathrm{O_{traj}}\|\mathbf{X})$ | *CarLLaVA* | Front image; speed of ego; GPS target point | $\emptyset$ | $\emptyset$ |
| $\mathbf{O_{traj}} \sim p(\mathrm{O_{traj}}\|\mathbf{X}, \mathbf{O_{text}})$ | *S4-Driver* | Multiview images; speed, acceleration and past trajectory of ego; high level command | What is my future trajectory? | Meta-decision, e.g., keep speed then do decelerating |
| $\mathbf{O_{traj}} \sim p(\mathrm{O_{traj}}\|\mathbf{X}, \mathbf{O_{text}}, \mathbf{O_{inittraj}})$ | *FeD* | Front image; speed of ego; GPS target point and high level command | Please evaluate the predicted future locations of ego vehicle | Collision with vehicles or pedestrians, traffic light violations, deviation from the expert and planned route |

### 4.2.1 Models without CoT Reasoning ❶

As one of the earlier approaches that uses a VLM for trajectory planning, **CarLLaVA** (Renz et al., 2024) is built on LLaVA-NeXt (Liu et al., 2024b), which combines ViT (Dosovitskiy et al., 2021) vision encoder with Tiny-Llama (Touvron et al., 2023a) as the LLM. Given the front-view image, current ego speed and two GPS target points as a route indicator, the model predicts (i) 20 *path waypoints* that are equidistant points at 1 meter apart, thus independent of the time, to control the steering and (ii) 10 *speed waypoints*, which are points at equal time intervals, specifically 0.25 seconds apart, to control the accelerator and the brake. These outputs are obtained in one-pass using MLPs as the planning head, following the design in Fig. 6(c)(Right). Note, unlike other methods we explore, the Tiny-Llama in CarLLaVA is trained from scratch. In 2024, this model, trained on approximately 3 million images, won the CARLA 2.0 challenge (Dosovitskiy et al., 2017), a competition based on driving in a closed-loop simulator. Given three camera views as front, front-left and front-right, **DriveGPT4-v2** (Xu et al., 2025c), as a similar model, combines SigLIP-ViT (Zhai et al., 2023b) with Qwen (Bai et al., 2023a) to predict the target speed and turning angle controls directly. In addition to that, the model is supervised to predict a trajectory and equidistant *route waypoints*, similar to the path waypoints of CarLLaVA. Different from CarLLaVA, *an expert model provided with privileged information*, such as the objects in the scene and potential hazard information, is trained to augment the training dataset of DriveGPT4-v2. This augmentation is shown to have a notable effect on driving performance. While these two approaches do not have a system prompt $\mathbf{T}$, **V2X-VLM** (You et al., 2024) uses a fixed $\mathbf{T} =$ *"Please predict the ego vehicle positions over next 45 timestamps."*. V2X-VLM is built on Florence-2 (Xiao et al., 2023) (using DaViT image encoder (Ding et al., 2022) with BART language model (Lewis et al., 2020)), and utilises DAIR-V2X dataset (Yu et al., 2022) to focus specifically on *leveraging an external camera* from the infrastructure in the environment.

> **Summary of trade-offs:** *Models without CoT Reasoning* ❶
>
> As these approaches are solely designed for driving, the dataset curation is relatively simple, where each data example typically consists of $(\mathbf{X}, \mathbf{O_{traj}})$ pairs. Moreover, due to the absence of CoT reasoning, these models demonstrate superior computational efficiency particularly when trajectory generation can be accomplished through a single forward pass. Nevertheless, these advantages entail certain trade-offs: the omission of CoT reasoning may limit driving performance, and the lack of language generation capabilities diminishes their explainability.

### 4.2.2 Text Output for CoT Reasoning ❷

CoT reasoning is typically a prompting strategy of the FMs that help them to break down the problem into multiple steps during inference. This is commonly achieved by prepending a step-by-step solution to a similar problem (Wei et al., 2022; Yao et al., 2023; Sel et al., 2024) to the input prompt, as a demonstration on how to approach and solve the problem. Furthermore, simply appending *"Let's think step by step"* to the prompt is also found to be quite effective, also known as the *zero-shot CoT* (Kojima et al., 2022). These enhanced ways of prompting strategies facilitate FMs to yield its output step-by-step, improving their performance in certain tasks such as arithmetic reasoning.

Similarly, FMs tailored for trajectory planning leverage CoT reasoning for improving the driving performance, however, in a slightly nuanced manner. In particular, these approaches explicitly *fine-tune the FMs* to yield the output step-by-step, instead of modifying the input prompt $\mathbf{T}$ during inference time. Besides, the prompt $\mathbf{T}$, in this case, is typically fixed, e.g., *"What is my future trajectory?"* (see S4-Driver in Tab. 2), and can potentially involve multiple steps. As a result of the fine-tuning, text output $\mathbf{O_{text}}$ includes a predefined set of explanations that are expected to improve trajectory planning, i.e., $\mathbf{O_{traj}}$. The scope of $\mathbf{O_{text}}$ varies significantly across methods, ranging from only a meta-action to a comprehensive description of the scene and agents' behaviours (please refer to Fig. 2(b) for a more comprehensive example).

Methods using a more comprehensive $\mathbf{O_{text}}$ as the CoT reasoning generally follow the common pipeline of the modular AD models (see Fig. 3(a)), hence generating the CoT reasoning in the order of perception and prediction information followed by planning decision, e.g, meta-action and trajectory. As a pioneering approach following this sequence, **GPT-Driver** (Mao et al., 2023) fine-tunes Chat-GPT3.5 (OpenAI, 2023) to yield notable objects in the scene, along with their potential impact on ego, and a meta-action as CoT reasoning before yielding the proposed trajectory. As an earlier method, GPT-Driver adopts a *text interface* as the vision adapter (refer to Fig. 6(b)(Left)), hence it is not E2E-trainable. Specifically, the perception and prediction information is extracted by the corresponding modules of UniAD (Hu et al., 2023b), and propagated to Chat-GPT3.5 as text. **DriveVLM** (Tian et al., 2024) follows a similar sequence in terms of CoT reasoning, where scene description and analysis are followed by meta-action prediction, and finally trajectory planning. *To preserve the capabilities* that the VLM gained before fine-tuning, Qwen-VL (Bai et al., 2023b) is fine-tuned by combining AD datasets with the LLaVA dataset, which includes examples from different domains, referred to as *co-tuning*. Similar to the previous methods, **Auto-VLA** (Zhou et al., 2025b) also relies on a comprehensive $\mathbf{O_{text}}$ for CoT reasoning, including a scene description, identification of the crucial objects, the intentions of the surrounding agents, and ideal driving actions. Differently, it follows *an adaptive CoT mechanism* where the model refers to CoT only for complex scenarios, considering its inefficient nature. To train such a model, a dataset is constructed as a combination of action-only scenes and reasoning-augmented scenes. The model is first trained using supervised fine-tuning, followed by reinforcement learning, specifically GRPO (Shao et al., 2024b), where the reward function promotes driving-related measures such as safety and comfort while discouraging CoT reasoning if deemed unnecessary.

There are also methods that generate a brief description or explanation as CoT reasoning. One example is **RAG-Driver** (Yuan et al., 2024), which only predicts an explanation of the action that the ego is executing, such as *"The car moves forward then comes to a stop because traffic has stopped in front"*, known as action explanation and justification in BDD-X dataset (Kim et al., 2018). Relying on this as CoT reasoning, the model outputs speed and turning angle of the ego. As the model is named after, it leverages *retrieval-augmented generation (RAG)*, which retrieves relevant information from external sources to enhance the model's prediction. Accordingly, in the AD domain, given a test sample, the two most similar samples in terms of cosine similarity are retrieved from the training data and fed to the LLM. As the retrieved data have known control signals, they are expected to improve driving performance for the test sample. Alternatively, **S4-Driver** (Xie et al., 2025) show that it is even useful to predict only one of the four pre-determined meta-actions (i.e., {keep stationary, keep speed, accelerate and decelerate}) via CoT reasoning before planning the trajectory. The model is built on PaLI-3 VLM (Chen et al., 2023b), which is based on the UL2 LLM (Tay et al., 2022), and equipped with custom modules to facilitate learning 3D scene representations.

**The inference time and accuracy trade-off of CoT reasoning.** One key impact of CoT reasoning that needs to be taken into consideration is its effect on inference time. This is because $\mathbf{O_{text}}$, as the CoT reasoning, is typically generated in an AR manner, requiring multiple passes through the LLM. This is especially important for the applications requiring fast inference speed, such as trajectory planning. To provide an example, we test the throughput of SimLingo, a relatively lightweight model with 1B parameters that processes only the front image, with and without CoT reasoning. Specifically for SimLingo, $\mathbf{T} = $ *"Predict the waypoints."* prompt yields the trajectory without CoT reasoning, while using $\mathbf{T} = $ *"What should the ego do next?"* triggers a relatively lightweight CoT reasoning with 2 or 3 short sentences about the what the action should be and why. For each of these prompts, we propagate 200 images to SimLingo five times using its official implementation. We observe that the CoT reasoning increases the inference time of SimLingo by $4.5\times$, from 3.6fps to 0.8fps on an Nvidia A4500 GPU, which can be a significant bottleneck for practical deployment. In this case, the

authors report a relatively small performance gain of using CoT reasoning, where the driving score increased from 84.41 to 85.07 and success rate improved from 64.84 to 67.27 on Bench2Drive benchmark (Jia et al., 2024). As a result, the compute requirement of the designed CoT reasoning as well as the performance gain it contributes to should be taken into consideration while using it in practice.

> **Summary of trade-offs:** *Models using Text Output for CoT* ❷
>
> While text-based CoT reasoning offers the advantage of grounding trajectories in the model's rationale, thereby enhancing explainability and potentially driving performance, these benefits come with substantial trade-offs. Foremost, the additional requirement of annotating training examples with text output $\mathbf{O_{text}}$ increases the complexity of data curation. Furthermore, as previously established, CoT reasoning can drastically increase inference latency, thereby imposing severe constraints on the practical deployment of such models in real-world applications.

### 4.2.3 Initial Trajectory Prediction for CoT Reasoning ❸

Different from using only $\mathbf{O_{text}}$ for the CoT reasoning, few approaches leverage it to assess or refine a trajectory that the model initially predicted as $\mathbf{O_{inittraj}}$. Such methods can also leverage $\mathbf{O_{text}}$ to improve $\mathbf{O_{inittraj}}$ or $\mathbf{O_{traj}}$ further. As an earlier approach, **Agent-driver** (Mao et al., 2024) essentially extends GPT-Driver (Mao et al., 2023) (Sec. 4.2.2) with a multi-step CoT reasoning involving—notable objects (with their potential effects) and a meta-action ($\mathbf{O_{text}}$), and an initial trajectory $\mathbf{O_{inittraj}}$. A predicted occupancy map from the perception module is also used to check whether $\mathbf{O_{inittraj}}$ *resulted in a collision.* If so, $\mathbf{O_{inittraj}}$ is corrected as $\mathbf{O_{traj}}$, otherwise it is accepted as the final trajectory $\mathbf{O_{traj}}$. Following GPT-Driver, it is built on GPT 3.5 and employs a text interface as the vision adapter, hence it is not E2E-trainable. Alternatively, **FeD** (Zhang et al., 2024) first predicts a trajectory $\mathbf{O_{inittraj}} \sim p(\mathrm{O_{traj}}|\mathbf{X})$ with the initial system prompt *"Predict ten future locations in 2.5 seconds"*. This is then followed by refining $\mathbf{O_{inittraj}}$ using *the feedback of the same model as the CoT reasoning* including potential collisions, traffic light violations or deviations from the route or expert behaviour, as detailed in Tab. 2. Finally, the trajectory is refined, i.e., $\mathbf{O_{traj}} \sim p(\mathrm{O_{traj}}|\mathbf{X}, \mathbf{O_{text}}, \mathbf{O_{inittraj}})$ by conditioning on the feedback as $\mathbf{O_{text}}$ and the initial trajectory $\mathbf{O_{inittraj}}$. As a result, unlike Agent-driver, relying on a frozen occupancy prediction model, *the feedback on* $\mathbf{O_{inittraj}}$ *is learned* in an E2E manner during training, and provided as an output such that $\mathbf{O_{traj}}$ considers it. FeD also employs an expert model with privileged information, similar to DriveGPT4-v2, to augment the dataset with the demonstrations from the expert (refer to Sec. 4.2.1), but differently by using feature distillation from the expert. Alternatively, **Solve-VLM** (Chen et al., 2025) is designed to have the final trajectory always in two-steps using *Trajectory CoT*. Specifically, a coarse trajectory, $\mathbf{O_{inittraj}}$, is predicted among a set of predetermined trajectories, and then, $\mathbf{O_{inittraj}}$ is refined to be more precise as $\mathbf{O_{traj}}$. Unlike FeD, the model does not explicitly make explanations, e.g., about potential collisions or traffic violations. Solve-VLM uses a custom VLM by combining EVA-02 (Fang et al., 2024) with a LLaVA-based LLM via their proposed SQ-Former to incorporate 3D representations into the vision features before passing them to the LLM. As Solve (Chen et al., 2025) also transfers knowledge to another AD model, we will further explore it in Sec. 5 while discussing methods leveraging knowledge transfer.

> **Summary of trade-offs:** *Models using Initial Trajectory Prediction for CoT* ❸
>
> These approaches offer the advantage of iterative trajectory refinement, thereby potentially enhancing driving performance. By utilising exclusively the initial trajectory prediction for CoT reasoning (i.e., without incorporating additional text-based CoT), these methods can be adopted with computational efficiency, particularly when trajectories are generated through a single forward pass. Given that these models perform self-refinement of their outputs, the data curation does not require an additional overhead; however, the model design must incorporate appropriate mechanisms to facilitate this refinement process. A notable drawback, though, is that without text generation capabilities, these models lack direct explainability and cannot interact with the user.

Table 3: Characteristics of the models providing language interaction capability. If a model is trained in multiple settings such as DriveLM-Agent or Emma, then we include only one of them for clarity.

| Model | Training Dataset | Training Dataset Size | Overview of Language Annotations | Main Annotator for Language | Augmentation of Q&As with FM | An Example Question | Model Training | Language Evaluation |
|---|---|---|---|---|---|---|---|---|
| *DriveGPT4* (Xu et al., 2024) | BDD-X +Custom Q&As +Additional non-AD datasets | 16K BDD-X Q&As are enhanced with 40K Chat-GPT Q&As as the custom dataset | Action description, action justification, scene description | Human for BDD-X, and Chat-GPT for the custom | Questions of BDD-X | Is there any risk to the ego vehicle? | Step 1. Vision-language alignment by only training vision adapter Step 2. Mix-fine-tuning by combining both AD and non-AD datasets | CIDEr, BLEU, ROUGE, GPT-Score |
| *DriveLM-Agent* (Sima et al., 2024) | DriveLM-nuScenes | 4K front view images with 300K Q&A pairs | Video clip level scene descriptions. Image-level perception, prediction, planning, behaviour and motion VQAs | Humans | Questions | What object should the ego vehicle notice first / second / third when the ego vehicle is getting to the next possible location? | Fine-tuning using the training set | SPICE and GPT-Score for perception, planning and prediction VQAs, and classification accuracy for behaviour VQA. |
| *Emma* (Hwang et al., 2025) | Custom Dataset +Waymo Open Motion Dataset | Custom Dataset: 203K hours of driving Waymo Open Motion Dataset: 572 hours of driving | 3D object detection, drivable road graph estimation, road blockage estimation | Using perception annotations labeled by humans | ✗ | Is the road ahead temporarily blocked? | Step 1. Pretraining on the large-scale custom dataset Step 2. Fine-tuning using Waymo Open Motion Dataset | Task metrics such as Precision-Recall curve for object detection |
| *OpenDriveVLA* (Zhou et al., 2026) | nuCaption +nuScenes-QA +nu-X | nuScenes dataset with ~4 hours of driving scenarios in 700 video clips | nuCaption: Description of the scene, objects and potential risks nuScenes-QA: Scene understanding Q&As including existence, counting, query-object, query-status and comparison-type questions nu-X: Action description and justification | nuCaption: LLaMA-Adapter and GPT-4 nuScenes-QA: Perception annotations by humans nu-X: Human | nuCaption: Both questions and answers nuScenes-QA: None nu-X: Only answers | Are there any cars to the front right of the stopped bus? | Step 1. Vision-language alignment Step 2. Fine-tuning for VQAs Step 3. Fine-tuning for motion prediction Step 4. Fine-tuning for trajectory planning | nuCaption: BLEU and BERT-Score nuScenes-QA: Accuracy nu-X: CIDEr, BLEU, METEOR and ROUGE |
| *DiMA-MLLM* (Hegde et al., 2025) | DriveLM-nuScenes extended with the remaining images in nuScenes | nuScenes dataset with ~4 hours of driving scenarios in 700 video clips | Video-clip level scene descriptions. Image-level perception, prediction, behaviour and motion VQAs | Human, and Llama-3-70B for the extension | No additional augmentation for the extension | What are the future movements of the agents to the back right of the ego car? | Step 1. Training only the E2E model Step 2. Fine-tuning E2E model and VLM jointly | Qualitative evaluation |
| *Omni-Q/L* (Wang et al., 2025b) | OmniDrive | nuScenes dataset with 700 video clips for training (~4 hours of driving scenarios) | Scene descriptions, attention, counterfactual reasoning (as in the example question), decision making and planning, general conversation | GPT-4 | ✗ | If I decide to accelerate and make a left turn, what could be the consequences? | Step 1. Vision-language alignment using 2D perception tasks Step 2. Fine-tuning the model | CIDEr for language evaluation, Average Precision and Average Recall for counterfactual reasoning |
| *Orion* (Fu et al., 2025a) | Chat-B2D | B2D-Base dataset with ~7 hours of driving scenarios and 2.11M Q&As in 950 video clips | Scene description, behaviour description of critical objects, meta-actions and action reasoning of the ego, recall of essential historical information | Based on CARLA simulator state and Qwen2VL-72B | ✗ | How has the current speed changed compared to the previous frames? | Step 1. Vision-language alignment using 2D perception tasks Step 2. Language-action alignment without VQA dataset Step 3. Fine-tuning the model with VQA dataset | Qualitative evaluation |
| *SimLingo* (Renz et al., 2025) | Custom | 3.1M front images are annotated in DriveLM style | Image-level perception, prediction, planning, behaviour and motion VQAs | Based on CARLA simulator state | Questions and Answers | Is there a traffic light in the scene? | Fine-tuning using the training set | SPICE and GPT score on DriveLM dataset |

## 4.3 Models Providing Additional Capabilities

We now discuss models that provide language or action capabilities in addition to trajectory planning.[2]

### 4.3.1 Models Providing Language Interaction Capability ❹

Clearly, the ability to interact via language is a remarkable feature enabled by the language interface of LLMs. Tab. 3 provides an overview of approaches using language interaction, including characteristics regarding their training datasets, training approaches, and language evaluations. Below, we summarise our observations regarding their training datasets and evaluations. **Language Interaction Capability Datasets.** Models with language interaction capability generally require pairs of VQAs for training, which are not part of standard driving datasets such as nuScenes (Caesar et al., 2020). Hence, after creating the necessary question templates, there are three main approaches to annotate these datasets, depending on whether the dataset is real-world or simulated:

- For simulation datasets, usually curated using CARLA (Dosovitskiy et al., 2017), predefined answer templates are populated by the *simulation state* including agents' velocities and positions, the weather, junctions, and traffic lights. This approach is used in DriveLM-CARLA (Dosovitskiy et al., 2017) and the SimLingo dataset (Renz et al., 2025), and is a relatively easy approach resulting in accurate language annotations.

---

[2]Since SimLingo is the only approach providing both capabilities ❻ , we present its language capabilities in Tab. 3 and discuss SimLingo in detail in Sec. 4.3.2, instead of reserving a section for it.

- For real-world datasets, the desired details of the scene are usually not available. As a result, some approaches rely on *human annotations* such as DriveLM-nuScenes (Sima et al., 2024). Though this results in accurate annotations, it requires significant resources to annotate a large dataset.
- Alternatively, FMs such as GPT-4 (OpenAI, 2024) are frequently used to create language labels. Specifically, the question templates are provided to the FM along with a system prompt and sensor data to obtain the answers. This approach is used to annotate both simulation datasets (Fu et al., 2025a) and real-world datasets (Hegde et al., 2025). As this is an *automated approach*, it is efficient, though manual inspection might be necessary to ensure the quality of the annotations.

After constructing the VQA dataset, several methods *augment* questions or answers using an LLM to increase language variability. Furthermore, the size of the training dataset varies significantly across methods. For instance, while DriveGPT4 is trained on 56K VQAs, Orion uses 2.11M VQAs for training. Additionally, the VQAs generally focus on perception, prediction and planning as subtasks that allow a model to achieve good driving performance. DriveLM (Sima et al., 2024) and OmniDrive (Wang et al., 2025b) are proposed as VQA benchmarks. The DriveLM dataset includes VQA pairs for these tasks, such as *"what object should the ego vehicle notice first when the ego vehicle is getting to the next possible location?"*. Alternatively, OmniDrive stands out with *counterfactual VQAs*, such as *"If I decide to accelerate and make a left turn, what could be the consequences?"*, where the counterfactual questions are designed using templates relying on the meta-actions such as *accelerate and make a left turn* in this example. Consequently, the answers are obtained by a combination of a rule-based checklist and GPT-4.

**Evaluating Language Interaction Capability.** For evaluating the quality of the generated text, the methods commonly employ measures from natural language processing literature, comparing machine-generated text with reference text. These measures include BLEU (Papineni et al., 2002), ROUGE (Lin, 2004), METEOR (Banerjee & Lavie, 2005) as well as CIDEr (Vedantam et al., 2015) and SPICE (Anderson et al., 2016), which are specifically introduced for image captioning. Furthermore, model-based evaluation measures are also utilised. For example, GPT-Score (Fu et al., 2023), used by Sima et al. (2024), is obtained by prompting Chat-GPT with the question, the reference answer, and the machine-generated answer, and asking for a numerical score about the accuracy of the machine-generated answer. Alternatively, Zhou et al. (2026) use BERT-Score (Zhang* et al., 2020) to compare the similarity of the machine-generated text with the reference in the BERT embedding space (Devlin et al., 2019).

**A Discussion of Existing Approaches.** Here, we highlight the key aspects of the trajectory planning approaches providing language capability. Please refer to Tab. 1 and Tab. 3 for the details on their model design and language interaction capability, respectively. **DriveGPT4** (Xu et al., 2024) aims to retain the capabilities of the LLM while training a trajectory planning model. To achieve this, they found it useful to *keep non-AD related VQAs* in addition to 56K driving-related VQAs during training. As the model is based on a custom VLM obtained by combining a video encoder (Luo et al., 2025) with Llama2 (Touvron et al., 2023b), the training of the model is handled in two stages, vision-language alignment and model fine-tuning, following Fig. 6(d)(Right). Alternatively, **DriveLM-Agent** (Sima et al., 2024), the proposed baseline of the DriveLM benchmark, fine-tunes BLIP-2 (Li et al., 2023a), a VLM based on the Flan-T5 LLM (Chung et al., 2024), on the VQAs of the benchmark in a single stage. **Emma** (Hwang et al., 2025), based on Gemini VLM (Gemini Team, 2025), uses a large dataset of 203K hours of driving scenes for the *pretraining* of the model. This pretrained model is then fine-tuned on specific domains for adaptation, such as on NuScenes dataset (Caesar et al., 2020). Instead of the conversation type questions and answers, the language capabilities of Emma seems to be limited to the questions regarding a predetermined set of perception tasks including object detection, road graph estimation, and road blockage detection. Different from the existing approaches, **OpenDriveVLA** (Zhou et al., 2026) and **DiMA-MLLM** (Hegde et al., 2025) rely on BevFormer (Li et al., 2025b) as the vision encoder to *effectively extract a 3D representation of the scene* to address the limitation of the FM-based vision encoder such as CLIP (Radford et al., 2021; Zhai et al., 2023b) or EVA (Fang et al., 2024). However, this can also cause a potential disadvantage in comparison to such models, considering that BevFormer is not pretrained on a large dataset. **Omni-Q** and **Omni-L** (Wang et al., 2025b) share the same architecture, where EVA-02-L (Fang et al., 2024) vision encoder is combined with LLaVA-v1.5 LLM (Liu et al., 2024a), *except their vision adapters*. Specifically, Omni-L relies on a linear layer following LLaVA (Liu et al., 2023) (see Fig. 6(b)(Middle)) while Omni-Q employs a Q-Former

Table 4: Characteristics of the models providing action interaction capability.

| Model | Training Dataset (with sensors and size) | Example Notice Instructions | Example Action Instructions | Mechanism to Avoid Misleading Instructions | Model Training | Instruction-following Evaluation Measures |
|---|---|---|---|---|---|---|
| *DriveMLM* (Cui et al., 2025a) | 280 hours of driving scenarios in CARLA. Sensors: 4 cameras (front, rear, left and right) and a lidar sensor. | ✗ | • I'm running short on time. Is it possible for you to utilise the emergency lane to bypass the vehicles ahead?
• Great view on the left. Can you change to the left lane?
• There are obstacles ahead. Can you switch to a different lane to bypass? | ✗ | Fine-tune the VLM using the training dataset | Qualitative evaluation |
| *LMDrive* (Shao et al., 2024a) | 3M driving scenarios collected in CARLA at 10fps (∼83 hours of driving data). Sensors: 4 cameras (front, rear, left and right) and a lidar sensor | • Please watch out for the pedestrians up ahead.
• Be mindful of the vehicle crossing on a red light to your left.
• Please be alert of the uneven road surface in the vicinity ahead. | • Feel free to start driving.
• Find your way out at the first exit on the roundabout, please.
• At the forthcoming T-intersection, execute a right turn. Just head for the left lane. Maintain your course along this route. | ✓ | Step 1. Train vision encoder with perception tasks using the front image
Step 2. The model is trained end-to-end | Driving performance is estimated while the model is provided action and notice instructions. LangAuto benchmark is proposed for this purpose in the same paper. |
| *SimLingo* (Renz et al., 2025) | 3.1M front images (∼215 hours of driving data) collected in CARLA at 4fps | ✗ | • Gently press the brakes.
• Hit the vehicle Ford Crown.
• Direct one lane to the left. | ✓ | Fine-tune the VLM using the training dataset | Accuracy of the model for each type of action interaction |

supervised by 3D perception tasks (refer to Fig. 6(b)(Right)). Their analyses suggest that using a *linear layer is more beneficial than a Q-Former* in terms of both language capabilities and open-loop driving performance. Finally, **Orion** (Fu et al., 2025a) combines EVA-02-L with Vicuna-1.5 (Vicuna Team, 2023) using a Q-Former variant. The proposed Q-Former, coined as QT-Former, also considers the temporal aspect of the observations through a memory bank to improve the driving performance. Furthermore, Orion presents notable analyses on using different architectures as planning heads (refer to Fig. 6(c)(Right)). Their analyses conclude that *the planning head based on a variational autoencoder* perform better than using an MLP or a diffusion model.

---

**Summary of trade-offs:** *Models Providing Language Interaction Capability* ❹

This group of approaches offers the distinct advantage of answering user questions, primarily serving to enhance model explainability and provide reassurance regarding the system's decision-making processes (i.e., trajectory). Nevertheless, similar to text-based CoT reasoning, the generation of text output through an AR mechanism can substantially increase inference time, thereby constraining the deployment of this functionality. Furthermore, dataset curation for these approaches presents greater complexity compared to methods only using text-based CoT. This is because developing language interaction capabilities typically requires each observation annotated with multiple question-answer pairs, in contrast to the singular text output per training example requisite for text-based CoT reasoning.

---

### 4.3.2 Models Providing Action Interaction Capability ❺

We now review models that can plan a trajectory by considering instruction from the user. We present an overview of these methods in terms of training dataset, training approach and evaluation in Tab. 4.

**Action Interaction Capability Datasets.** All three approaches in this category (refer to Fig. 4) are trained and tested on *synthetic datasets* collected using CARLA simulator (Dosovitskiy et al., 2017). This is likely because annotating data with pairs of instructions and actions is relatively easier for synthetic data as the simulator state includes comprehensive information about the scene. Furthermore, unlike other approaches discussed in this paper, both DriveMLM and LMDrive rely on both lidar and camera. Additionally, LMDrive stands out with the functionality to consider *notice instructions* such as *watch the tunnel coming up.* Shao

et al. (2024a) introduce a specific benchmark called **LangAuto-Notice** to measure the performance of the model to respond to such notice instructions, demonstrating that LMDrive effectively leverages notice information to improve driving performance. Another crucial capability that such models are expected to have is a mechanism to avoid *misleading instructions*, as such instructions can result in unsafe consequences. Ideally, upon capturing a misleading instruction, the model should reject it and follow a safe trajectory. Both LMDrive and SimLingo incorporate this crucial capability into their models.

**Evaluating Action Interaction Capability.** Unlike language interaction, the evaluation measures for action interaction are *not yet well-established*. In addition to the qualitative evaluation, which is limited in terms of the number of examples and tends to have a selection bias, two different approaches are adopted to evaluate action interaction. Shao et al. (2024a) design a benchmark in which the model is instructed solely by the user, similar to an extended version of the navigational commands, where the standard CARLA performance measures are reported. As an alternative, Renz et al. (2025) directly measure the *percentage of the trajectories that is following the given instruction*, namely, the accuracy of the model to follow instructions.

**A Discussion of Existing Approaches.** Tab. 4 provides an overview of the approaches with action interaction capability. **DriveMLM** (Cui et al., 2025a) combines EVA (Fang et al., 2023) and GD-MAE (Yang et al., 2023) as vision and lidar encoders, respectively, with Llama LLM (Touvron et al., 2023a) through Q-Former modules. The model is then supervised to output four predefined commands (i.e., {keep, accelerate, decelerate, stop}) for controlling the accelerator, and five steering actions (i.e, {follow, left change, right change, left borrow, right borrow}). Similarly, **LMDrive** (Shao et al., 2024a) processes inputs from multiple cameras and lidar using a multimodal vision encoder including ResNet (He et al., 2016) and PointPillars (Lang et al., 2019). This encoder is initially pretrained using object detection, traffic light status classification, and trajectory planning before it is combined with Llama (Touvron et al., 2023a) where the model is supervised for trajectory planning. Unlike other approaches, the LLM is kept frozen during the training. To determine if *the given user instruction is completed*, the model predicts an additional flag, making LMDrive the only approach with this functionality. Finally, **SimLingo** (Renz et al., 2025) is built on Mini-InternVL VLM (Gao et al., 2024c) as an extension to CarLLaVA (Sec. 4.2). Different from other approaches, SimLingo has ❻ *both language and action interaction capabilities*, as well as the option to use CoT reasoning (please refer to our discussion on inference time-accuracy trade-off of CoT reasoning in Sec. 4.2.2 for further details). The model is trained in a single stage on a dataset including training examples for these functionalities. The instruction following capability, coined as *action dreaming*, aims to align the predicted trajectory with the natural language instructions. Consequently, it is shown that action dreaming helps improving the driving performance more than the pure language tasks, i.e., VQA and CoT reasoning, as shown in the paper. SimLingo also pay attention to the resampling of the dataset via carefully creating data buckets for predefined driving characteristics and then assign different sampling ratios to each bucket.

> **Summary of trade-offs:** *Models Providing Action Interaction Capability* ❺
>
> The approaches with only action interaction capability offer the distinct advantage of trajectory planning conditioned on user instructions, serving as a potentially valuable additional feature. Since this functionality itself does not require generating text output, such models can be deployed efficiently in real-world systems. However, the dataset curation protocols for developing these models remain insufficiently investigated for real-world systems. Essentially, the model requires training with a substantial volume of text input (user instructions) paired with corresponding output trajectories. Crucially, the text input should exhibit sufficient variability to enable robust responses to diverse user instructions. Akin to other subgroups lacking text generation, these models could inherently suffer from a deficiency in explainability especially when CoT reasoning is not employed either.

## 5 Foundation Models Guiding Trajectory Planning

An alternative way of utilising FMs for trajectory planning is to use an existing approach for AD, such as a modular or an E2E one (see Fig. 3(a,b)), and transfer the knowledge of a chosen FM into it either only during training or during both training and inference. A key distinction among these approaches come from their choice of using FM during inference which, of course, results in increased computational cost and

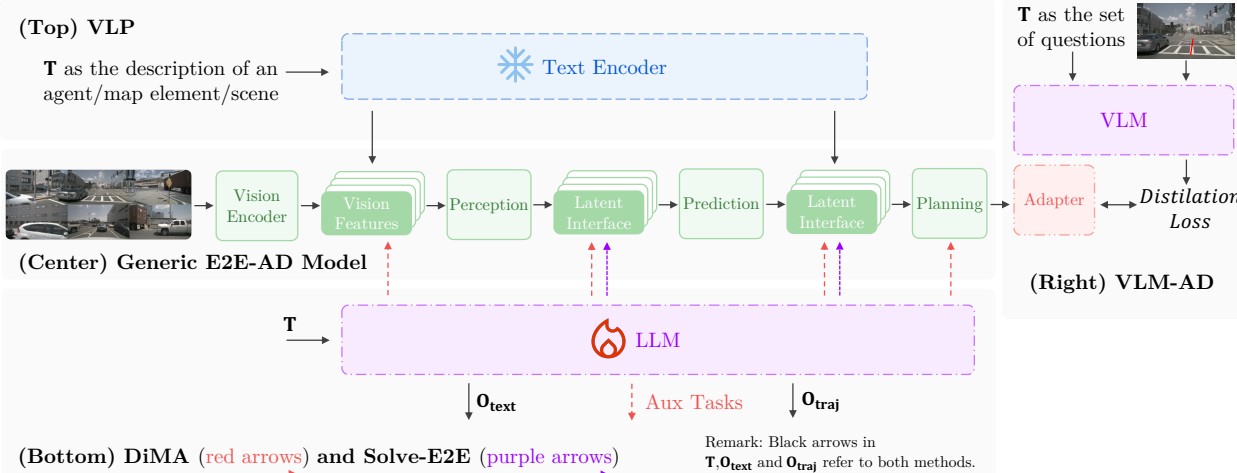

Figure 7: Overview of the approaches that uses an FM only during training for knowledge distillation. **(Center)** A generic E2E-AD model (please refer to Sec. 2 for the details.) **(Top)** VLP prompts the model with the agent/map elements/scene descriptions to align the representations of CLIP text encoder with those of the E2E-AD model. The arrows from the text encoder represent the representations. **(Right)** VLM-AD prompts a VLM with the privileged information (the red line on the input image) and a few questions to align the representations of the E2E-AD model. **(Bottom)** DiMA and Solve jointly trains E2E-AD model and the FM, where the LLM makes predictions by conditioning on the representations of the E2E-AD model. Please refer to Sec. 5.1 for further details.

memory requirements. Considering that, based on our taxonomy (refer to Fig. 4), we discuss the models using knowledge distillation only during training in Sec. 5.1, and those requiring an FM also for inference in Sec. 5.2.

## 5.1 Models Using Knowledge Distillation Only During Training 🛈

As this group of approaches carries out knowledge distillation only during training, they do not need the FM for inference. Fig. 7 provides an overview of the existing approaches within this category. Among the first approaches, **VLP** (Pan et al., 2024a) aims to align the representations of an E2E-AD model with an off-the-shelf CLIP (Radford et al., 2021). To obtain CLIP representations, only the text encoder of the CLIP is used. Specifically, the text encoder is prompted by (i) the descriptions of each agent and map elements such as their semantic class and location, and (ii) a planning prompt, such as *"the self-driving car is driving in an urban area. It is going straight. Its planned trajectory is $\mathbf{O_{traj}}$"*. Then, these representations are distilled into the E2E-AD model in two levels as shown in Fig. 7(Top), using contrastive learning. While the agent and map element representations are distilled into vision features of the E2E-AD model, the representation of the planning prompt is incorporated into its ego query as the input to the planning module. Similar to VLP, **VLM-AD** (Xu et al., 2025b) keeps the FM frozen but instead prompts a VLM, GPT-4o (OpenAI, 2024), with (i) the front image including the privileged information of the future trajectory of the ego shown by the red line on the image in Fig. 7(Right), and (ii) six different questions to obtain a meta-action and its reasoning from the VLM. These responses are then structured using one-hot encoding and a CLIP text encoder, and an adapter layer is appended to the E2E-AD model to predict this structured output. Finally, cross-entropy loss is used for knowledge distillation.

Different from the previous approaches, **DiMA** (Hegde et al., 2025) and **Solve** (Chen et al., 2025) train E2E-AD and LLM jointly for knowledge distillation. Specifically, as shown in Fig. 7(Bottom), both approaches propagate representations of the E2E-AD model to the LLM of LLaVA-v1.5 (Liu et al., 2024a), such that the LLM predicts the trajectory and a text output by conditioning on these representations. While Solve propagates only the representations after perception, DiMA aims for a more comprehensive distillation. As a result, in DiMA, (i) BEV features, agent/map representations, and the ego query are all propagated to the

Table 5: Design choices of the models that transfer knowledge from an FM during inference. ❄: Frozen, 🔥: Standard fine-tuning, 🔥: LoRA fine-tuning. Broadly speaking, the last three columns correspond to the feature volumes (latent interfaces in blue) in Fig. 3(b). For *FasionAD++*, the used FM type for the main experiments is not clear in the paper, hence it is marked as not available (N/A).

| | Model | Sensor Input to FM ($X_{FM}$) | Used FM | Transferred Knowledge from FM at Inference (Z) | How to Encode Z | Z Transferred Into | | |
|---|---|---|---|---|---|---|---|---|
| | | | | | | Vision Features | Prediction Input | Planning Input |
| 1 | *VLM-E2E* (Liu et al., 2025) | Front camera | BLIP-2 ❄ | Scene description | CLIP text encoder + MLP | ✓ | | |
| 2 | *DME-Driver* (Han et al., 2025) | Front camera | LLaVA 🔥 | Scene description, driver's gaze, and driver's logic | BERT | | ✓ | ✓ |
| 3 | *Senna-E2E* (Jiang et al., 2024) | 360° cameras | ViT-L + Vicuna-v1.5 🔥 | Meta-action | Learnable embedding layer | | | ✓ |
| 4 | *DiffVLA* (Jiang et al., 2025a) | 360° cameras | ViT-L + Vicuna-v1.5 🔥 | Meta-action | One-hot encoding | | | ✓ |
| 5 | *DriveVLM-Dual* (Tian et al., 2024) | Front camera | Qwen-VL 🔥 | Trajectory | None | | | ✓ |
| 6 | *Solve-E2E-Async* (Chen et al., 2025) | 360° camera features | LLaVA-1.5-LLM 🔥 | Trajectory | None | | | ✓ |
| 7 | *DiMA-Dual* (Hegde et al., 2025) | 360° camera features | LLaVA-1.5-LLM 🔥 | LLM planning features in the last layer | None | | | ✓ |
| 8 | *HE-Drive* Wang et al. (2024a) | 360° cameras | Llama3 🔥 | Weights of a scoring function to select the most suitable trajectory among multiple ones | None | | | ✓ |
| 9 | *VDT-Auto* (Guo et al., 2025) | Front camera | Qwen2-VL 🔥 | VLM features of the detected objects, meta-action, trajectory proposals | None | | | ✓ |
| 10 | *FasionAD++* (Qian et al., 2025b) | Front camera | N/A | Existence of predetermined objects (e.g., traffic lights, intersections, obstacles) and meta-actions | Binary encoding for object existence, learnable embeddings for meta-actions | ✓ | ✓ | ✓ |
| 11 | *AsyncDriver* (Qian et al., 2025b) | Vectorised encoding of the scene | Llama2 🔥 | Ego states, the occupancy of the adjacent scene, the state of the traffic light, lane change and velocity decisions | Learnable embedding layer | | | ✓ |

LLM (red arrows from LLM to the blue interfaces in Fig. 7(Bottom)), and (ii) a distillation loss is introduced to align the representations of the LLM and the planning module of the E2E-AD (red arrow from the LLM to the planning module in Fig. 7(Bottom)). Furthermore, DiMA includes auxiliary tasks such as masked BEV token reconstruction for effective representation learning during training. Both approaches fine-tune the LLM using LoRA (Hu et al., 2022) for the trajectory planning task.

> **Summary of trade-offs:** *Models Using Knowledge Distillation Only During Training* ❼
>
> These approaches are unique in this review by their elimination of FM dependency during inference. Consequently, the resulting AD models typically have fewer parameters and exhibit a higher inference rate, thereby facilitating the deployment of computationally efficient systems. However, this independence from FMs excludes access to natural language interaction capabilities. Specifically, such models cannot process user questions or instructions, and are constrained in terms of model explainability. Moreover, the complexity of the dataset curation process varies depending on whether the FM is jointly trained and the specific type of knowledge being distilled, which often necessitates additional input-output pair annotations.

## 5.2 Models Using Knowledge Transfer During Inference ❽

Unlike the methods in the previous section, the approaches we discuss here are the ones employing FMs during both training and inference for more effective knowledge transfer. Tab. 5 provides the main characteristics of these approaches, including the type of knowledge transferred from the FM and how this knowledge is integrated into the AD model. In the following, we elaborate on these approaches based on the type of knowledge each transfers to the AD model, which is usually a (i) scene description involving perception or prediction features, (ii) a planning decision such as a meta-action or trajectory, or (iii) both.

**Methods that Transfer Scene Description.** These methods aim to transfer scene descriptions, such as *"a black van driving in the ego lane, away from the ego car"*, taken from **VLM-E2E** (Liu et al., 2025).

Specifically, Liu et al. (2025) prompt an off-the-shelf BLIP-2 (Li et al., 2023a) with a front image of a driving scene to obtain such a description. As this description is essentially a text, CLIP text encoder (Radford et al., 2021) converts it into a text representation. Then, the representation is mapped to shifting and scaling factors using MLPs to update the BEV features of the E2E-AD model, in this case UniAD (Hu et al., 2023b). **DME-Driver** (Han et al., 2025) follows a similar approach, including knowledge of the scene description as well as *the driver's gaze* and the driver's logic on the taken action, such as the presence of the pedestrians for slowing down. Particularly, given the front view of the driving scene, LLaVA (Liu et al., 2023) is fine-tuned to produce the text output, which is converted into embeddings using BERT (Devlin et al., 2019). Finally, BEV features of the E2E-AD model attend to these embeddings through cross-attention before being fed into the occupancy prediction and planning modules of UniAD (Hu et al., 2023b). Though these methods show that such a knowledge transfer is useful, they do not guide the E2E-AD model *explicitly* with a planning decision such as a meta-action or a trajectory, which we discuss next.

**Methods that Transfer Planning Decisions.** These methods transfer the planning decisions of the FM to the AD model, usually in the form of a meta-action or trajectory. For example, **Senna-E2E** (Jiang et al., 2024) transfers the meta-action of the VLM by fine-tuning it specifically for this task. The model is fine-tuned in multiple stages to *progressively* specialise the VLM for planning: In driving fine-tuning, the VLM is supervised with VQAs in driving scenarios, followed by planning fine-tuning for meta-action classification. After the VLM is trained, it is frozen and *the meta-action of the VLM is converted into an embedding* using a learnable layer. Then, in order to benefit from the knowledge of the VLM, the planning query in the AD model, VADv2 (Jiang et al., 2026), attends to this meta-action embedding of the VLM. **DiffVLA** (Jiang et al., 2025a) also relies on Senna-VLM but with two main differences. First, instead of a learnable embedding layer, a one-hot encoding of the meta-action is passed to the planning module. Second, the VADv2 planner is replaced by a *diffusion planner* (Liao et al., 2025), which generates the trajectory by conditioning on this meta-action encoding as well as the BEV features, map and object queries *sequentially*. The approach is trained and evaluated using the NAVSIM v2 benchmark (Cao et al., 2025b).

Differently, some approaches, including DriveVLM, Solve and DiMA, fine-tune their VLMs for the trajectory planning task. This manifests itself as an additional advantage of combining the planned trajectories from the AD model and the VLM for potentially improving the driving performance. To begin with, **DriveVLM-Dual** (Tian et al., 2024) uses the trajectory of the VLM as *the input query of the planning module* of the E2E-AD model, such that the planning module refines it further. This approach is incorporated into UniAD (Hu et al., 2023b), VAD (Jiang et al., 2023) and AD-MLP (Zhai et al., 2023a). Similarly, **Solve-E2E-Async** (Chen et al., 2025) uses the trajectory of the LLM as an additional query of the planning module in the E2E-AD model, yet *asynchronously*. That is, to account for the longer inference time of the VLM, (i) the prediction horizon of the VLM is designed to be longer than that of the E2E-AD model, and (ii) the E2E-AD model uses the last predicted trajectory of the VLM as the additional planning query. Alternatively, **DiMA-Dual** (Hegde et al., 2025) transfers *the representation yielded by VLM for trajectory planning*. Specifically, max-pooling is used on the last layer features of the planning representations obtained in the VLM and E2E-AD models, and then the E2E-AD model predicts the trajectory from the resulting pooled features. Finally, as a different approach, **HE-Drive** (Wang et al., 2024a) fine-tunes Llama3 (Llama Team, Meta AI, 2024) to output the weights of a function scoring candidate trajectories. Specifically, Wang et al. (2024a) use a diffusion-based planner to sample multiple candidate trajectories, each of which is scored considering various driving-related factors such as collision risk, target speed compliance and comfort. For example, if there is a stopped vehicle in front of ego and the ego needs to slow down, the weight of the target speed compliance is increased by the VLM, thereby helping to select the best trajectory.

**Methods that Transfer Scene Description and Planning Decision.** Some approaches transfer both a scene description and a planning decision. **VDT-Auto** (Guo et al., 2025) fine-tunes Qwen2-VL (Bai et al., 2023b) on driving-related VQAs to output object descriptions, as well as a meta-action and trajectory. The corresponding representations in the VLM are then transferred to the AD model by freezing the VLM. Specifically, these embeddings are used as inputs to a *diffusion-based planner* (Liao et al., 2025), which refines the trajectories conditioned on these embeddings and the vision features. Alternatively, **FasionAD++** (Qian et al., 2025b) guides the VLM to output (i) a planning state including binary variables indicating the existence of, e.g., traffic lights, obstacles, intersections; and (ii) a meta-action. The planning state updates BEV

features, agent, map and ego queries through an adapter layer, while the meta-action is incorporated into the ego features only. As a result, FasionAD++ provides knowledge transfer at multiple levels into the E2E-AD model. Additionally, FasionAD++ takes the inefficiency of the VLM into account by referring to the VLM only when the *uncertainty of the AD model* increases beyond a certain threshold. Unlike other methods that we discuss in this section, **AsyncDriver** (Chen et al., 2024c) transfers LLM knowledge to the planning module of a modular approach (see Fig. 3(a)). Specifically, it fine-tunes an LLM to predict useful information for AD by appending it with an *assistance alignment module.* Conditioned on the output of the LLM, this module predicts relevant perception features such as traffic light states, occupancy of the adjacent lane, as well as features affecting the trajectory planning such as lane change and velocity decisions. The latent encoding of the LLM, used as the input to the assistance alignment module, is propagated to the modular planner via a feature adapter. To align the LLM and the modular planner, they are trained jointly on the nuPlan dataset (Caesar et al., 2022). During inference, the latent encoding from the LLM is passed *asynchronously* to the modular planner to improve the planning decision, maintaining the previous high-level instruction features during intervals. This asynchronous connection between the separate real-time and LLM-based planners provides a balance between quality and inference speed for responses.

> **Summary of trade-offs:** *Models Using Knowledge Transfer During Inference* ❽
>
> This category of approaches encompasses systems that integrate a FM with an AD model. The inference latency of such coupled architectures relies upon specific design parameters. For instance, continuous reliance on the FM by the AD model can substantially increase inference overhead, whereas selective and conditional FM invocation can enhance average computational efficiency. Additionally, the inference time is influenced by whether the transferred knowledge is obtained following AR steps within the FM. Regarding explainability, the integrated FM component might generate natural language explanations; however, ensuring semantic alignment between the AD model's decision-making processes and the FM's textual outputs presents considerable challenges. Similar to approaches employing knowledge distillation, the complexity of dataset curation depends upon whether the FM undergoes joint training and the specific nature of the information being transferred, necessitating the collection of additional text input-output pairs.

## 6 How Open Are Data and Code of the Existing Approaches?

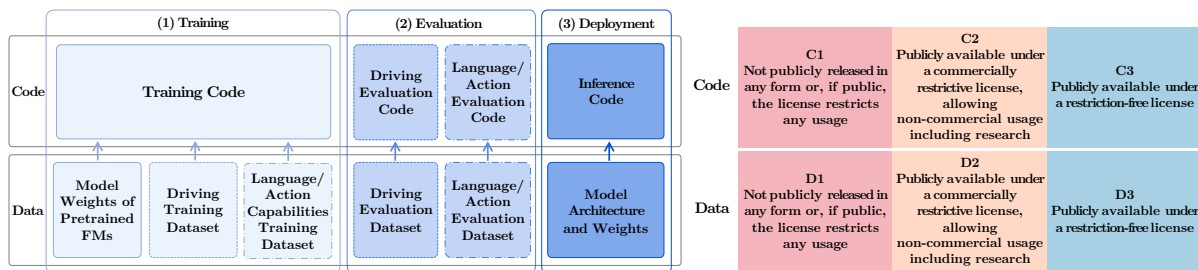

(a) The code and data for each stage of development     (b) The levels of openness for each asset

Figure 8: Representation of different stages of model development, and the levels of openness.

Open models and datasets play a vital role in the advancement of research and acceleration of practical deployment. They enable researchers to quickly build upon existing work to improve the state-of-the-art, while allowing practitioners to build high-performing real-world systems without the extensive time needed to reproduce the existing work. It also fosters user trust as transparency reveals strengths and weaknesses of approaches. Moreover, open-source resources lower economic barriers, promoting adaptation in low-income regions. Considering the importance of openness, we present the level of openness for all the 37 approaches for trajectory planning considered in this work.

Table 6: Openness characteristics. While every effort has been made to ensure the accuracy of this table, we cannot guarantee its completeness or correctness. The information is provided for general reference only and should not be interpreted as legal advice. If in doubt, please confirm with the corresponding authors. The table reflects the status as of 08 October 2025. Entries marked as N/A indicate that the asset is "not applicable" to the method. We further outline the conventions we used while classifying the openness of the approaches in Appendix A, which also clarifies the symbols * and **. We label the first method of each taxonomy group with its corresponding icon.

| Method | Training | | | | Evaluation | | | | Deployment | |
|---|---|---|---|---|---|---|---|---|---|---|
| | Training Code | Model Weights of Pretrained FMs | Driving Training Dataset | Language/Action Capabilities Training Dataset | Driving Evaluation Code | Driving Evaluation Dataset | Language/Action Evaluation Code | Language/Action Evaluation Dataset | Inference Code | Model Architecture and Weights |
| 1 CarLLaVA (Renz et al., 2024) | C3 Apache2.0 | D3 LLaVA-NeXT | D2 Custom | N/A | C3 Apache2.0 | D3 Bench2Drive | N/A | N/A | C3 Apache2.0 | D1 |
| 2 DriveGPT4-v2 (Xu et al., 2025c) | C1 | D3 Qwen-0.5B | D1 Custom | N/A | C1 | D3 CARLA Longest6 | N/A | N/A | C1 | D1 |
| 3 V2X-VLM (You et al., 2024) | C1 | D3 Florence-2 | D3 DAIR-V2X | N/A | C1 | D3 DAIR-V2X | N/A | N/A | C1 | D1 |
| 4 GPT-Driver (Mao et al., 2023) | C1 No license | D1 GPT-3.5 | D2 nuScenes | N/A | C1 No license | D2 nuScenes | N/A | N/A | C1 No license | D1 |
| 5 Drive-VLM (Tian et al., 2024) | C1 | D3 Qwen-VL | D2 nuScenes | N/A | C1 | D2 nuScenes | N/A | N/A | C1 | D1 |
| 6 Auto-VLA (Zhou et al., 2025b) | C1 | D3 Qwen2.5-VL | D2 WOMD | N/A | C1 | D3 Bench2Drive | N/A | N/A | C1 | D1 |
| 7 RAG-Driver (Yuan et al., 2024) | C3 Apache2.0 | D2 Vicuna v1.5 | D2 BDD-X | N/A | C3 Apache2.0 | D2 BDD-X | N/A | N/A | C3 Apache2.0 | D1 |
| 8 S4-Driver (Xie et al., 2025) | C1 | D1 PaLI-3 | D2 WOMD | N/A | C1 | D2 nuScenes | N/A | N/A | C1 | D1 |
| 9 Agent-driver (Mao et al., 2024) | C3 MIT | D1 GPT-3.5 | D2 nuScenes | N/A | C3 MIT | D2 nuScenes | N/A | N/A | C3 MIT | D1 |
| 10 FeD (Zhang et al., 2024) | C1 | D2 LLaVA-7B | D3 CARLA | N/A | C1 | D3 LAV | N/A | N/A | C1 | D1 |
| 11 Solve-VLM (Chen et al., 2025) | C1 | D2 LLaVA v1.5 | D2 nuScenes | N/A | C1 | D2 nuScenes | N/A | N/A | C1 | D1 |
| 12 DriveGPT4 (Xu et al., 2024) | C1 | D2 LLaMA2 | D2 BDD-X | D1 Custom, no license | C1 No license | D2 BDD-X | C1 No license | D1 Custom, no license | C1 No license | D1 |
| 13 DriveLM-Agent (Sima et al., 2024) | C1 | D3 BLIP-2 | D2 DriveLM | D2 DriveLM | C3 Apache2.0 | D2 DriveLM | C3 Apache2.0 | D2 DriveLM | C1 | D1 |
| 14 Emma (Hwang et al., 2025) | C1 | D1 Gemini | D1 Custom | D1 Custom | C1 | D2 nuScenes | C1 | D2 WOD | C1 | D1 |
| 15 OpenDriveVLA (Zhou et al., 2026) | C1 | D3 Qwen2.5 | D2 nuScenes | D1 nu-X, no license | C3 Apache2.0 | D2 nuScenes | C3 Apache2.0 | D2 nuScenes-QA | C3 Apache2.0 | D1 |
| 16 DiMA-MLLM (Hegde et al., 2025) | C1 | D2 LLaVA v1.5 | D2 nuScenes | D1 Custom | C1 | D2 nuScenes | C1 | D2 DriveLM | C1 | D1 |
| 17 Omni-Q (Wang et al., 2025b) | C2 Custom | D2 LLaVA v1.5 | D2 nuScenes | D2 Custom | C3 Custom | D2 nuScenes | C3 Custom | D2 DriveLM | C2 Custom | D3* |
| 18 Omni-L (Wang et al., 2025b) | C2 Custom | D2 LLaVA v1.5 | D2 nuScenes | D2 Custom | C3 Custom | D2 nuScenes | C3 Custom | D2 DriveLM | C2 Custom | D1 |
| 19 Orion (Fu et al., 2025a) | C3 Apache2.0 | D2 Vicuna v1.5 | D3 B2D-Base | D3 B2D-Chat | C3 Apache2.0 | D3 Bench2Drive | C3 Apache2.0 | D3 Bench2Drive | C3 Apache2.0 | D3* |
| 20 DriveMLM (Cui et al., 2025a) | C1 | D2 LLaMA-7B | D1 Custom | D1 Custom | C1 | D3 CARLA Town05 | C1 | D1 Custom | C1 | D1 |
| 21 LMDrive (Shao et al., 2024a) | C3 Apache2.0 | D2 LLaVA v1.5 | D2 Custom | D2 Custom | C3 Apache2.0 | D3 CARLA Town05 | C3 Apache2.0 | D2 Custom | C3 Apache2.0 | D2** |
| 22 SimLingo (Renz et al., 2025) | C3 Apache2.0 | D3 MiniInternVL | D2 Custom | D2 Custom | C3 Apache2.0 | D3 Bench2Drive | C3 Apache2.0 | D2 DriveLM | C3 Apache2.0 | D3 |
| 23 VLP (Pan et al., 2024a) | C1 | D3 CLIP | D2 nuScenes | N/A | C1 | D2 nuScenes | N/A | N/A | C1 | D1 |
| 24 VLM-AD (Xu et al., 2025b) | C1 | D1 GPT-4o | D2 nuScenes | N/A | C1 | D3 CARLA Town05 | N/A | N/A | C1 | D1 |
| 25 DiMA (Hegde et al., 2025) | C1 | D2 LLaVA v1.5 | D2 nuScenes | N/A | C1 | D2 nuScenes | N/A | N/A | C1 | D1 |
| 26 Solve-E2E (Chen et al., 2025) | C1 | D2 LLaVA v1.5 | D2 nuScenes | N/A | C1 | D2 nuScenes | N/A | N/A | C1 | D1 |
| 27 VLM-E2E (Liu et al., 2025) | C1 | D3 BLIP-2 | D2 nuScenes | N/A | C1 | D3 CARLA Town05 | N/A | N/A | C1 | D1 |
| 28 DME-Driver (Han et al., 2025) | C1 | D2 LLaVA | D1 Custom | N/A | C1 | D2 nuScenes | N/A | N/A | C1 | D1 |
| 29 Senna-E2E (Jiang et al., 2024) | C3 Apache2.0 | D2 Vicuna v1.5 | D1 DriveX | N/A | C3 Apache2.0 | D2 nuScenes | N/A | N/A | C3 Apache2.0 | D2** |
| 30 DiffVLA (Jiang et al., 2025a) | C1 | D2 LLaVA v1.5 | D2 NAVSIM v2 | N/A | C1 | D2 NAVSIM v2 | N/A | N/A | C1 | D1 |
| 31 DriveVLM-Dual (Tian et al., 2024) | C1 | D3 Qwen-VL | D1 Custom | N/A | C1 | D2 nuScenes | N/A | N/A | C1 | Model |
| 32 Solve-E2E-Async (Chen et al., 2025) | C1 | D2 LLaVA v1.5 | D2 nuScenes | N/A | C1 | D2 nuScenes | N/A | N/A | C1 | D1 |
| 33 DiMA-Dual (Hegde et al., 2025) | C1 | D2 LLaVA v1.5 | D2 nuScenes | N/A | C1 | D2 nuScenes | N/A | N/A | C1 | D1 |
| 34 HE-Drive (Wang et al., 2024a) | C1 | D2 LLaMA 3 | D2 nuScenes | N/A | C1 | D2 nuScenes | N/A | N/A | C1 | D1 |
| 35 VDT-Auto (Guo et al., 2025) | C1 | D3 Qwen-VL | D2 nuScenes | N/A | C1 | D2 nuScenes | N/A | N/A | C1 | D1 |
| 36 FASIONAD++ (Qian et al., 2025b) | C1 | Unavailable | D2 nuScenes | N/A | C1 | D3 Bench2Drive | N/A | N/A | C1 | D1 |
| 37 AsyncDriver (Chen et al., 2024c) | C3 Apache2.0 | D2 LLaMA 2 | D2 nuPlan | N/A | C3 Apache2.0 | D2 nuPlan | N/A | N/A | C3 Apache2.0 | D3** |

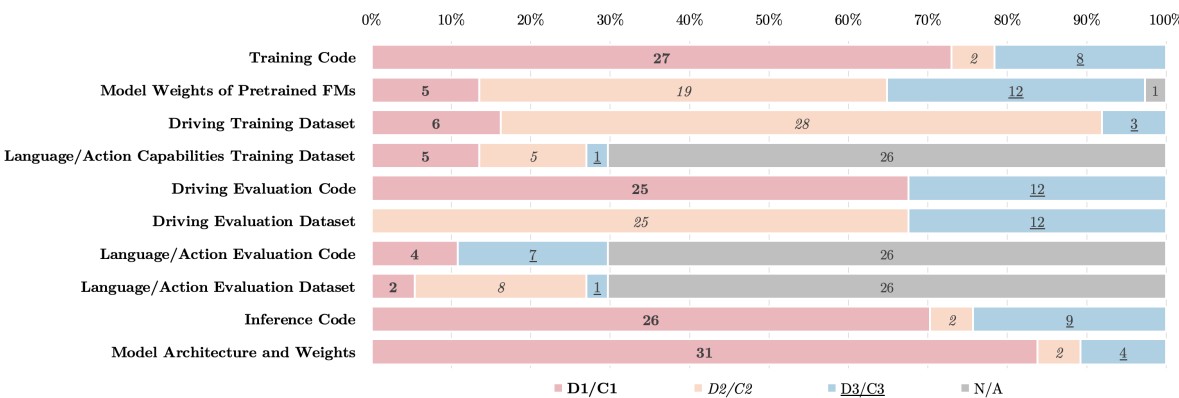

Figure 9: Distribution of openness levels across data and code assets for the 37 surveyed methods. Numerical labels on the bars represent the count of approaches in each openness category per asset.

Following Eiras et al. (2024), we consider training, evaluation and deployment as different stages of building a model, and accordingly score the assets, which is "data" and "code". For an asset, a higher score implies a higher openness level. Since we study trajectory planning methods that leverage FMs, the underlying assets for our use cases require additional elements, as shown in Fig. 8(a). Specifically, we include *pretrained FM weights* in the training stage as an additional data asset. We also consider driving and language/action datasets separately, as language annotations are generally derived from existing driving datasets, and multiple language annotations can exist for the same driving dataset, e.g., nuCaption (Yang et al., 2025b), nuX (Ding et al., 2024a), nuScenes-QA (Qian et al., 2024) and DriveLM-nuScenes (Sima et al., 2024). For the levels of openness, we score each asset between 1 and 3 to provide targeted information for research and commercial purposes (refer to Fig. 8(b)), defined as:

- 1 implies an asset, i.e., code or data, is not publicly released or is released with a license that restricts both non-commercial and commercial usage, including research.
- 2 represents that the asset is publicly available and can be used for non-commercial purposes, including research, but not for commercial use. One example license for this is Creative Commons Attribution-NonCommercial 4.0 International.
- 3 is for assets that are publicly available and can be used for both research and commercial purposes, e.g., those with Apache License v2.0 or MIT License.

In Tab. 6, we classify the code and data of all the approaches using these levels of openness. We observe that *there is no approach with all assets being available for both research and commercial purposes.* Only five of the approaches (i.e., Omni-Q, Orion, LMDrive, SimLingo and AgentDriver) released all of their assets openly with some assets limited for commercial usage. Four of these approaches are FMs tailored for trajectory planning, and one of them is an FM guiding trajectory planning of a modular AD model. Consequently, *no open-source implementation is available* for FM guiding trajectory planning for an E2E-AD model (rows 23-36).

To demonstrate and summarise per-asset openness status, we provide the distribution of the openness levels of each asset in Fig. 9. The figure indicates that the model weights of pretrained FMs are commonly open for research (31 out of 37 models), enabling the models to be developed. On the other hand, this doesn't necessarily translate into their derivatives being open. Specifically, *the model architecture and weights is the most restricted asset with only 6 models providing them openly for research and commercial usage (D2 or D3).* Similarly, as another crucial aspect of reproducibility, *only 10 models openly share the training code (C2 or C3).* This discrepancy creates a significant *bottleneck*: without model parameters and implementation details of the training pipelines, it is difficult for the community to build upon the previous research. Ultimately, this presents a challenge for reproduction and reuse.

We also observe a clear distinction between the openness of synthetic and real-world datasets. Synthetic datasets, generated typically via the CARLA simulator (Dosovitskiy et al., 2017), are often fully-open (D3),

allowing for unrestricted use in both research and commercial projects. In contrast, the real-world datasets used by the surveyed methods, including nuScenes Caesar et al. (2020) and BDD-X Kim et al. (2018), are commonly restricted to non-commercial use cases only (D2). This gap creates friction when translating research into industrial practice and may introduce inconsistencies in standardised evaluation across academic and commercial domains.

## 7 Open Issues and Emerging Directions

In this section, we identify and discuss the open issues within our scope, while also considering emerging directions in the broader research landscape.

**Deploying these models can be challenging due to the high inference cost, especially when CoT reasoning is used.** Except for a few models using knowledge distillation only during training (Sec. 5.1), all other approaches in our scope require an FM for inference. Consequently, mainly due to the large number of parameters and AR generation of the output, their latency is commonly longer than what is suitable for AD. This is usually accepted as 10-30fps to provide effective responsiveness to the changing environment and to match the latency of sensor data (camera, lidar) (Caesar et al., 2020; Sun et al., 2020; Ettinger et al., 2021). For example, Orion (Fu et al., 2025a) has more than 7B parameters (see Tab. 1) and an inference rate of 0.8fps on an Nvidia A800 GPU, even without CoT (Fu et al., 2025b). Additionally, referring to our previous experiment in Sec. 4.2.2, the throughput of SimLingo (Renz et al., 2025), a relatively smaller model than its counterparts, using only the front view, without and with CoT reasoning, is 3.6fps and 0.8fps, respectively, on an Nvidia A4500 GPU respectively. While, efficient processing has been recently considered in a number of models, such as the use of separate slow, reasoning-based FM systems alongside a fast decision system for rapid response times (Qian et al., 2025b; Zhou et al., 2025b; Wang et al., 2025c), fast control cycles have generally not been a priority for current models. Therefore, more efficient approaches are necessary to be able deploy such models in the real-world.

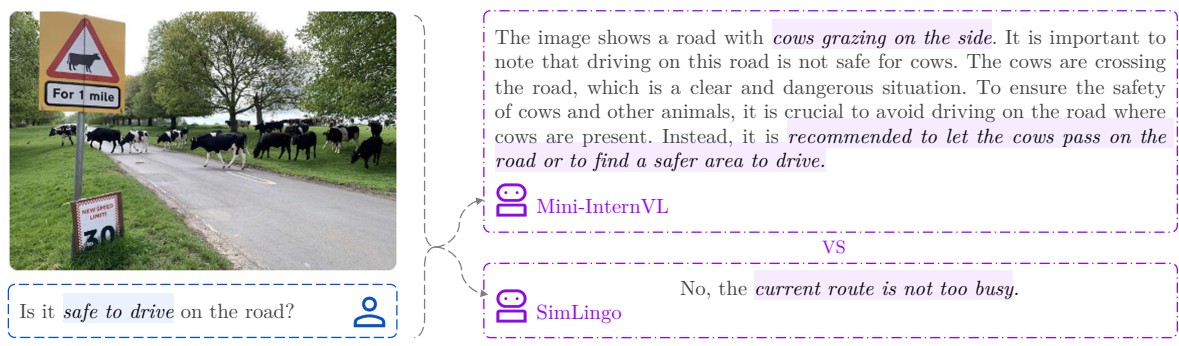

Figure 10: An example showing *vision-language collapse* in the VLM of SimLingo after fine-tuning. We first prompt Mini-InternVL, the VLM that SimLingo is initialised from, and then SimLingo, i.e., Mini-InternVL after fine-tuning, with the image and text shown in the figure.

**Fine-tuned VLMs become *less capable* of interpreting the world (vision-language collapse).** The superiority of FMs tailored for trajectory planning in benchmarks demonstrates that they effectively leverage transfer learning via fine-tuning the pretrained models. However, this fine-tuning procedure can result in the loss of capabilities of the VLM that could potentially be more helpful for trajectory planning. This crippling effect might be emerging due to the negative effect of fine-tuning, studied as *concept forgetting* (Mukhoti et al., 2024) which is an extension of the traditional catastrophic forgetting (Kirkpatrick et al., 2017) for fine-tuned FMs. We show this effect in Fig. 10, where we prompt the VLM used in SimLingo with the same image and text inputs before and after fine-tuning. For comparison purposes, we use one of the inputs we employed for prompting GPT-4o in Fig. 1, where GPT-4o provides a comprehensive accurate response. Although it is not as comprehensive as GPT-4o, the response of Mini-InternVL, a relatively small VLM that SimLingo is initialised from, is also quite accurate. Specifically, it captures that *the cows are crossing the*

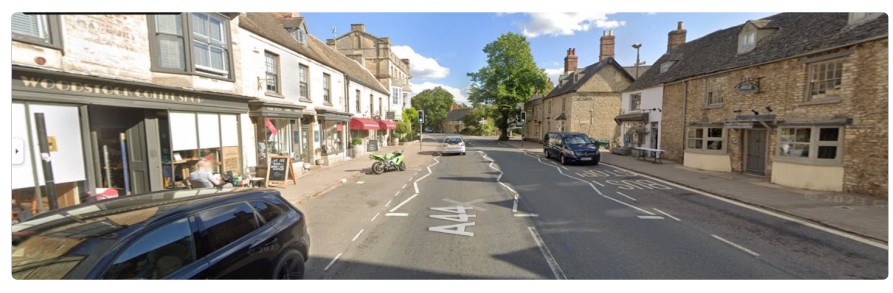

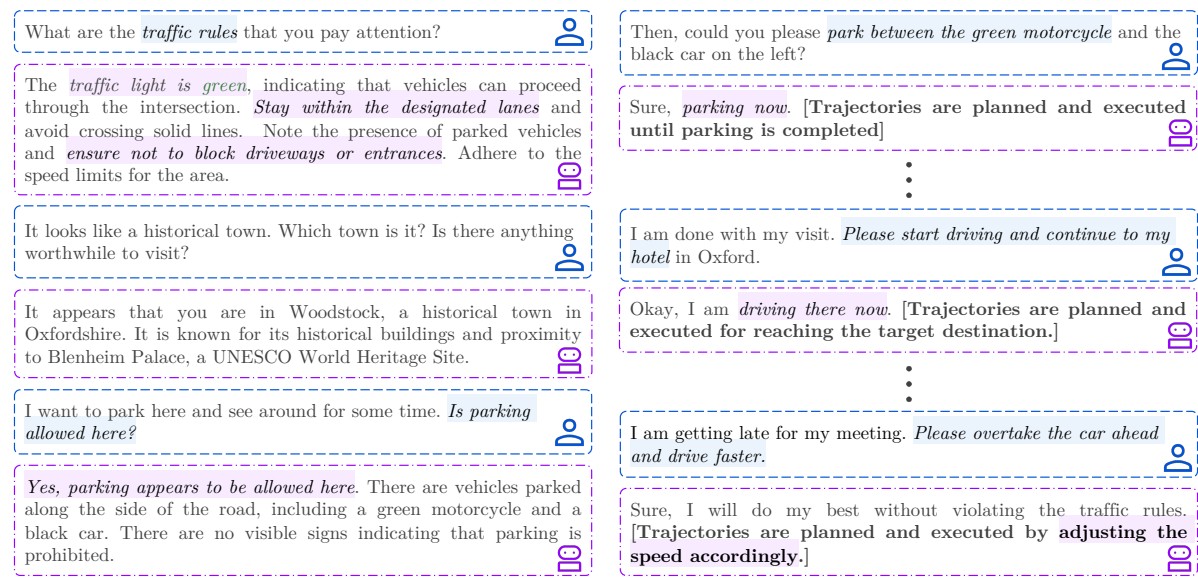

Figure 11: Example VQAs and user instructions that the current models lack.

*road* and suggests *either letting the cows pass or finding a safer area to drive.* On the other hand, after fine-tuning, SimLingo provides a *self-contradictory response*, mentioning that it is not safe to drive followed by an explanation that the route is not busy, implying a safe route. Furthermore, SimLingo does not provide any insight about the environment, such as the presence of the cows. Further research and thorough analysis is essential to ensure preservation of necessary capabilities of VLMs after fine-tuning on AD-specific datasets.

**Towards Agentic VLAs.** Existing approaches with language interaction capability generally focus on interpreting the current scene to provide explainability to the user. There is however potential to offer further use cases if the question-answer pairs are tailored accordingly. For example, the user can ask non-driving-related questions such as the historical details of a landmark in a town that the ego driver passes through, as illustrated in Fig. 11. Alternatively, the model could help people with visual impairments understand their surroundings, such as why the traffic is stopping. One limitation of current models with action interaction capability, is that they are limited to instructions that can be executed in a short time horizon. DriveMLM and SimLingo consider instructions to change speed or lane, which can be completed within the horizon of a single trajectory output. *As a result, the model does not need to remember the instruction over multiple calls, or check if it is completed, which is impractical.* Though the instruction set used in LMDrive is larger with 56 different types of instructions, it still doesn't require the model to break down a human-like complex instruction into steps as in the example *"please start driving and continue to my hotel in Oxford"*, illustrated in Fig. 11. More sophisticated planning models in the robotics literature (Huang et al., 2022; Belcak et al., 2025) can help leverage similar capabilities in real-world for AD models.

**Challenges of ensuring safety of action interaction capability in a real-world setup.** Providing action-interaction capability allows substantial flexibility of commands that can be provided by a user. However, due to the open domain of user instructions, there are substantial challenges in ensuring that the vehicle operates safety, regardless of the commands given. Existing methods do include training of action

safety in the dataset, as described in Section 3.1, however ensuring safety with an unrestricted command domain remains an open issue. Furthermore, all models investigated in this work with action interaction capability are trained and tested using synthetic setups using CARLA simulator. As domain shift results in a significant drop in performance (Luo et al., 2018; Zhou et al., 2023a; Oksuz et al., 2023; Kuzucu et al., 2024), deploying existing models without addressing sim-to-real gap is impractical. Therefore, designing methods to address sim-to-real gap and curating real-world training datasets is necessary to be able develop, and eventually deploy, models with action interaction capability.

**Further analyses are necessary to identify what is significant for improving driving performance.** There are major differences in model and experimental design choices across methods, which makes it difficult to understand why a method makes a difference. For example, Orion is trained on the B2D-Base dataset (Jia et al., 2024) with around 7 hours of video clips, combines an off-the-shelf vision encoder with a custom Q-Former with memory blocks for videos, trains the model in three stages, has more than 7B parameters, uses a variational autoencoder as trajectory planning head, and employs a comprehensive CoT including scene description, scene analysis, action reasoning, and history review. This models achieves 77.74 driving score on the closed-loop B2D benchmark, significantly outperforming E2E-AD approaches such as VAD with 42.35 driving score (Jia et al., 2024). On the other hand, SimLingo employs Mini-InternVL (1B parameters) as an off-the-shelf VLM, fine-tunes it on more than 200 hours of video in a single stage relying on a data resampling approach coined as bucketing, uses a different set of language annotations and CoT, and finally achieves a driving score exceeding 85. These major differences between models, training approaches as well as training datasets make it difficult to understand why a model makes a difference. As a result, a deeper understanding of how these factors affect the model is necessary.

**Lack of standardised benchmarks and metrics for FM-based trajectory planning.** Building upon the aforementioned challenges, the establishment of standardised benchmarks and metrics is essential to facilitate rigorous and equitable comparison across different approaches, and to systematically identify the key factors contributing to performance improvements (e.g., training data characteristics, architectural design choices, model scale). Such benchmarks and evaluation metrics should address multiple dimensions of model performance, including:

- *Driving performance under resource-constrained settings.* FM-based approaches for AD exhibit considerable variation in model size. For instance, SimLingo comprises 1B parameters (Renz et al., 2025), while Orion contains approximately 7B parameters (Fu et al., 2025a). Consequently, inference latency varies substantially across methods, further influenced by design decisions such as CoT usage. Therefore, evaluating driving performance under controlled resource constraints (e.g., bounded inference time) becomes crucial for assessing the practical viability of FM-based methods in real-world deployment scenarios.
- *Language interaction and action interaction capabilities.* Beyond core driving competencies, certain FMs designed for trajectory planning incorporate additional capabilities for language and action interactions. To comprehensively evaluate these capabilities, standardised benchmarks encompassing both real-world and simulated data are necessary.
- *Reasoning of the models.* Furthermore, analogous to the common benchmarks used for evaluating FMs, standard benchmarks for evaluating the reasoning capabilities of FMs tailored for trajectory planning are needed. Drawing from the broader FM literature, the ARC-AGI benchmark (Chollet et al., 2025) exemplifies this approach by designing visual reasoning tasks that humans solve intuitively while FMs struggle. Similar domain-specific benchmarks should be developed for AD to assess reasoning abilities when processing complex linguistic inputs, as well as multimodal language-vision grounding capabilities. Developing such benchmarks presents substantial challenges and warrants dedicated attention from the research community.

**Improving driving and evaluation with World Models.** Training current FM-based methods for AD typically relies on recorded driving demonstrations from an expert, which can be either human-generated or derived from another model with access to privileged information. Consequently, model performance is fundamentally bounded by the capabilities of the expert, and the availability of training data remains inherently limited. Furthermore, existing closed-loop simulators exhibit significant constraints: they are either characterised by low fidelity, as exemplified by CARLA (Dosovitskiy et al., 2017), or impose substantial

computational demands, as in the case of novel viewpoint synthesis approaches (Kerbl et al., 2023). In contrast, world models aim to *learn* the underlying dynamics of the autonomous vehicle's operating environment, thereby enabling the generation of plausible future states. Therefore, world models can be leveraged to synthesise novel training data (Ren et al., 2025; Zhao et al., 2025b) as well as to facilitate evaluation procedures that emphasise rare and safety-critical edge cases, thus addressing long-tail distributional challenges. Additionally, world models can also be integrated with driving models, allowing world dynamics to influence driving decisions (Wang et al., 2024d), for example by rolling out possible futures using the world model and selecting the most suitable trajectory for driving.

## 8 Conclusive Remarks

In this paper, we provided a comprehensive review of trajectory planning methods that utilise an FM. To offer a complete and coherent perspective, we introduced a taxonomy of these methods based on how an FM is employed. Using this taxonomy, we discussed the corresponding approaches separately in a detailed and comparative manner, providing a unified yet critical point of view. We also investigated the openness of the approaches to assist practitioners and researchers in selecting suitable models. Finally, we identified open issues that are critical for developing practical models with the desired functionalities. With this review, the community can better understand the current state of the field and the directions to pursue for developing more capable solutions for AD leveraging FMs.

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

## APPENDIX

## A  Further Details on the Openness of the Methods

We classify and discuss the openness of the assets of each method in Sec. 6. Here, we include the conventions, which also clarify few outlier cases in the openness classification in Tab. 6:

- If evaluation of an approach is performed on multiple datasets, the dataset with *the least restrictive license* is considered for the classifying "Driving Evaluation Dataset" and "Language/Action Evaluation Dataset" assets. The name of the dataset in question is noted as an explanation. This selection is chosen to reflect when at least part of the results can be reproduced by the community.

- On the other hand, if training is performed on multiple datasets jointly, the dataset with *the most restrictive license* is considered for classifying "Trajectory Planning Training Dataset" and "Language/Action Capabilities Training Dataset' assets. The name of the dataset in question is noted as an explanation. This convention is chosen as all datasets within a mixture are needed to reproduce the results. However, few methods, e.g. Drive-VLM (Tian et al., 2024), train the models on different dataset mixtures, with results reported independently. In such a case, the score of *the least restrictive* mixture of datasets is used for the corresponding cell entry.

- To provide further clarity on the basis of our classification, we include the license name for the code assets, and the dataset name for the data assets (except model weights).

- For Omni-Q (Wang et al., 2025b), Orion (Fu et al., 2025a) and AsyncDriver (Chen et al., 2024c), the license of the released model weights is in conflict with pretrained FMs weights. We classify these entries based on the released model license and warn the reader that additional restrictions may apply (noted as *).

- For LMDrive (Shao et al., 2024a) and Senna-E2E (Jiang et al., 2024), there are two contradicting licenses for the model weights, in the same huggingface space. We classify the entry based on the more restrictive license which also aligns with the openness score of the pretrained FMs weights (noted as **).

- If no license is specified, default copyright law applies, which is highly restrictive.

- If the model combines a vision encoder and an LLM–instead of using an off-the-shelf VLM–we consider the most restrictive license for the pretrained FMs weights classification. Note, any components trained from scratch are ignored (e.g. LLM in CarLLaVA Renz et al. (2024))

- Closed-loop evaluation benchmarks like CARLA Town05 (Prakash et al., 2021), CARLA Longest6 (Chitta et al., 2022), Bench2Drive (Jia et al., 2024) and LAV (Chen & Krähenbühl, 2022) are considered within the "Driving Evaluation Dataset" asset as they provide route splits or auxiliary scenario configuration files.

