# OpenReview forum: "Foundation Models for Trajectory Planning in Autonomous Driving: A Review of Progress and Open Challenges"
_TMLR — Accepted by TMLR_

### Review · Reviewer_t9if · 2026-01-07

**Summary Of Contributions:**

Summary of Contributions:

This paper provides a survey of foundation-model-based trajectory planning for autonomous driving. Its main contribution is a hierarchical taxonomy that organises existing methods into two broad classes: (i) foundation models tailored for trajectory planning, which are fine-tuned to directly predict trajectories, and (ii) foundation models guiding trajectory planning, which use FMs for knowledge distillation or auxiliary guidance within existing AD pipelines. Using this taxonomy, the paper systematically reviews 37 recent approaches, discussing their architectural choices, use of chain-of-thought reasoning, language/action interaction capabilities, and training–inference trade-offs. It also offers practical guidance on fine-tuning, data curation, and trajectory representations, evaluates code/data openness, and highlights key open challenges for future research.

Key Strengths:

* Well-structured taxonomy that brings order to a fragmented literature.
* Useful discussion of practical design trade-offs and reproducibility.

Key Weaknesses:

* Lacks quantitative benchmarking or direct empirical comparisons.
* Some discussions of challenges could be more concrete or actionable.

**Audience:**

Yes

**Audience Explanation:**

The paper would be of interest to at least a subset of TMLR’s audience, particularly researchers working on foundation models, autonomous driving, multimodal learning, and decision-making systems. By providing a structured taxonomy, critical synthesis of recent methods, and a focused discussion on design trade-offs and open challenges, the paper offers value both as a reference for newcomers to the area and as a conceptual framework for researchers actively developing FM-based trajectory planning methods.

**Broader Impact Concerns:**

I do not identify major unaddressed broader impact concerns specific to this work. The submission is a survey paper that analyzes and organizes existing research on foundation models for trajectory planning, and it does not introduce new algorithms, datasets, or deployment claims that would raise immediate ethical risks.
That said, the topic area—autonomous driving—does have inherent societal and safety implications, including concerns around robustness, accountability, bias, and safe deployment. These issues are already widely recognised in the field and are partially reflected in the paper’s discussion of open challenges (e.g., robustness, evaluation, and sim-to-real transfer). If desired, the paper could briefly acknowledge these broader implications in a dedicated paragraph, but the absence of a detailed Broader Impact Statement does not appear to be a significant omission given the survey nature of the work.
Overall, no additional Broader Impact requirements are strictly necessary for this submission.

**Claims And Evidence:**

Yes

**Claims Explanation:**

The claims of the paper are about taxonomy, categorisation, and synthesis of existing work, rather than new empirical results. These claims are supported by clear definitions, consistent use of the proposed taxonomy, and accurate citations to a broad set of representative methods. The paper presents its arguments clearly, avoids overclaiming, and appropriately frames limitations and open challenges, which is consistent with TMLR’s evaluation criteria for survey submissions.

**Requested Changes:**

1. The paper should more explicitly state that it does not aim to compare or rank methods empirically, but rather to organise and analyse them conceptually. While this is largely implicit, making it explicit early (e.g., in the Introduction or Scope subsection) would help avoid misinterpretation of the paper’s goals.

2. In a few cases, the boundaries between subcategories (e.g., CoT-based planning vs. language-interaction capabilities) could be clarified further to ensure that categorisation decisions are fully unambiguous and consistently motivated across all surveyed methods.

3. A compact table summarising key trade-offs across categories (e.g., inference cost, reliance on language supervision, deployment feasibility) would improve accessibility and practical usefulness.

4. While open challenges are discussed, the paper could more explicitly highlight the lack of standardised benchmarks and metrics for FM-based trajectory planning, and how this complicates fair comparison across approaches.

5. The paper could benefit from a short forward-looking subsection identifying emerging directions (e.g., world models, instruction-following safety, efficiency-aware FM integration) to further guide future research.

6. Some sections are dense due to the breadth of coverage; light tightening of language or additional signposting could further improve readability, especially for readers less familiar with autonomous driving pipelines.

---

> ### Author Response · Authors · 2026-01-29
> **Replies to Reviewer t9if**
>
> Thanks for the constructive feedbacks and appreciating our work. Following includes our replies and explanations regarding the changes in the revised draft based on the reviewer's feedback:
>
> **“1. the paper should more explicitly state that it does not aim to compare or rank methods empirically”:** We now include a new paragraph at the end of the Section 1.1 Scope (right after the contributions), where we specify how we selected the methods for this review. In that paragraph, we explicitly mention that “We emphasize that our objective is not to compare or rank these methods, but rather to organize and analyse them conceptually.”
>
> **“2. clarifying the boundaries between subcategories”:** Specifically, the set of the text inputs **T** that the models with language interaction capabilities is richer than the models providing text output as CoT, in which **T** typically does not exist or it can be limited to a single prompt. We have added a comparative discussion to clarify the difference between CoT-based planning vs. language-interaction capabilities in Sec. 3.1 (Models providing additional capabilities). In that section, we have also provided Orion and S4-Driver as characteristic examples of these groups to clarify the distinction better.
>
> **“3, missing discussion in taxonomy groups”:** Also considering the similar concern of the reviewer K1tz, we have included *“Summary of trade-offs" boxes as per-leaf-node summary* at the end each subsection where we discuss the approaches in Sections 4 and 5. In these boxes, we assess each of the subgroups from the perspectives of (i) data needs (including the reliance on language supervision), (ii) inference efficiency (also from the perspective of deployment feasibility), (iii) explainability and (iv) capabilities.
>
> **“4. explicitly highlight the lack of standardised benchmarks and metrics for FM-based trajectory planning”:** In the first submission, we had only considered Evaluating reasoning (the last paragraph of Section 7) as one of the missing benchmarks. In the revised manuscript, we have extended our discussion on evaluation to cover different evaluation aspects. Specifically, we now discuss the need for the standardised benchmarks and metrics for FM-based models from the perspectives of driving performance under resource-constrained settings, language interaction and action interaction capabilities and reasoning.
>
> **“5. identifying emerging directions”:** In the first submission, we had in fact considered some emerging directions, such as Agentic AI, under open issues. To make this more explicit, we have renamed Section 7 “Open Issues” as “Open Issues and Emerging Directions” and extended our discussions in this section also by considering the suggestions of the reviewer. Specifically, we have extended our discussions on (i) deployment challenges of the models due to the high inference cost, (ii) challenges of ensuring safety of action interaction capability in a real-world setup, (iii) the lack of benchmarks and metrics. Also, we have included a new paragraph to discuss the potential effect of world models, as another emerging direction.
>
> **“6. light tightening of language or additional signposting”:** We have improved the signposting by including (i) a summary of the trade-offs as boxes, (ii) coloured icons that represent each leaf node in our hierarchical taxonomy, and (iii) new bullet points in data curation paragraph of Sec.4.1.

---

### Review · Reviewer_K1tz · 2026-01-09

**Summary Of Contributions:**

# Summary

The authors provide a survey of recent approaches that use multi-modal foundation models (FMs) for trajectory planning in autonomous driving. The main contribution is a hierarchical taxonomy that separates (i) FMs tailored for trajectory planning (fine-tuned / adapted to directly output trajectories) and (ii) FMs guiding trajectory planning (where FM knowledge is transferred to a planner, e.g., via distillation), and connects these groups to minimal mathematical formulations (Sec. 3; Fig. 4–5; Eq. (1)–(2)). The paper claims to systematically analyze 37 methods under this taxonomy, and provides practical discussion on fine-tuning and data curation (Sec. 4.1). In addition, the authors include an 'openness' assessment (code/data/weights) with a structured scoring scheme (Sec. 6; Fig. 8; Tab. 6; Appendix A), and summarize open challenges such as deployment latency constraints, sim-to-real gaps, vision-language collapse after fine-tuning, and evaluation/benchmarking limitations (Sec. 7). Figure 1 reads as a motivating qualitative example of VLM-style scene understanding.

# Strengths and Weaknesses

## Strengths

* S1: The taxonomy is comprehensive and well structured. The two-level split (tailored vs guiding, and then further sub-branches) is intuitive and captures meaningful differences in how FMs are used in planning.
* S2: I like that the taxonomy is not only descriptive, but also linked to basic formal formulations (Sec. 3.1; Eq. (1)–(2); Fig. 5). The notation (observations, trajectories, text prompts/outputs) helps clarify the conditioning structure and makes the categorization more rigorous.
* S3: The technical overview is detailed and organized in a way that is useful for practitioners (Sec. 4–5; Tables 1–6). In particular, the discussion covers (although briefly) relevant dimensions such as data curation, model adaptation choices (e.g., vision adapters), trajectory representation alternatives, and training strategies (Sec. 4.1; Fig. 6).
* S4: The openness assessment is a clear contribution (Sec. 6; Tab. 6; Fig. 8; Appendix A). The asset-level scoring is explicit and useful for reproducibility, and it highlights that 'openness' is currently a bottleneck in this area.
* S5: The open-issues section is concrete and grounded in examples (Sec. 7). For instance, the paper discusses inference-time costs of reasoning-style generation (e.g., the reported CoT latency drop from 3.6 fps to 0.8 fps in one example) and provides an intuitive example.

## Weaknesses

* W1 (Diagram inconsistency): Figure 5 has a visual inconsistency with the mathematical formulation (Fig. 5; Sec. 3.1; Eq. (1)). For example, in Fig. 5(b) (Text as CoT), the formula indicates that the generated text output should condition the trajectory generation, but the diagram does not clearly show this dependency (missing/unclear input arrow / two-stage flow). Since Fig. 5 is central to the taxonomy, this affects clarity and technical correctness of the manuscript.
* W2 (taxonomy presentation too abstract): Sections 3.1 and 3.2 introduce the main taxonomy branches mostly at an abstract level (Sec. 3.1–3.2). Adding 1–2 representative method examples (with references) directly in these sections would improve readability and make the categorization easier to follow for readers not already familiar with the literature.
* W3 (Data curation without references): The 'Data curation' discussion is currently too abstract for a section positioned as practical guidance (Sec. 4.1). The paper would benefit from concrete dataset examples (with references).
* W4 (search or selection protocol missing): The claim of 'systematically analyzing 37 methods' is not fully traceable without an explicit selection protocol (Sec. 1–2). A short paragraph describing search sources/venues, time window, and inclusion/exclusion criteria would make the survey stronger and easier to update.
* W5 (missing discussion in taxonomy groups): While the paper categorizes methods well, it lacks a structured 'pros/cons/implications' synthesis per taxonomy group (Sec. 4–5 vs. Sec. 7). Right now, the open-issues section is cross-cutting, but the reader still has to infer 'when should I use which leaf node and why'. A per-leaf-node summary (strengths, limitations, compute/latency implications, data needs, typical failure modes) would significantly increase the value of the taxonomy.
* W6 (missing depth in asset availability): The openness analysis is good, but it could benefit from a clearer high-level synthesis of available datasets/assets and their characteristics (Sec. 6; Tab. 6).
* W7 (missing references): Some datasets which are mentioned in the manuscript are not properly cited (e.g. DAIR-V2X with "DAIR-V2X: A Large-Scale Dataset for Vehicle-Infrastructure Cooperative 3D Object Detection")

**Audience:**

Yes

**Audience Explanation:**

Yes. TMLR readers interested in foundation models, planning, and autonomy would benefit from (i) a structured taxonomy that organizes a rapidly growing literature, (ii) the practical consolidation of design dimensions (training strategies, trajectory representations, interaction modalities), and (iii) the asset-level openness audit that directly impacts reproducibility and reuse (Sec. 3–6). The open-issues section is also relevant to both researchers and practitioners because it frames constraints that matter for deployment and evaluation (Sec. 7).

**Broader Impact Concerns:**

No major broader impact concerns are raised by the survey itself, but given that autonomous driving is safety critical, a brief statement on dataset governance (privacy/licensing) and safety validation context would improve completeness and set appropriate expectations for deployment.

**Claims And Evidence:**

Yes

**Claims Explanation:**

Yes. The claims are supported by the provided manuscript, although it still needs some corrections before this manuscript can be published.

**Requested Changes:**

# Requested changes

## Critical

* C1: Fix Fig. 5 consistency: revise the diagrams (especially Fig. 5(b)) to match the conditioning / AR information flow implied by each respective formulation.
* C2: Add a brief survey methodology / selection protocol supporting the 'systematic analysis of 37 methods' framing (Sec. 1–2).
* C3: Add a per-taxonomy-group 'pros/cons/implications' synthesis (ideally a per-leaf-node summary table: strengths/limitations/risks/when-to-use, plus intuition on compute/latency and data needs).
* C4: Verify that all the mentioned datasets are properly cited (e.g. DAIR-V2X)

## Minor

* M1: Add representative method examples/citations directly in Sec. 3.1 and Sec. 3.2 when defining the taxonomy branches.
* M2: Strengthen Sec. 4.1 data curation with concrete dataset examples.
* M3: Improve Tab. 6 readability and summarize key assets in the main text.

---

> ### Author Response · Authors · 2026-01-29
> **Replies to Reviewer K1tz**
>
> Thanks for the constructive feedbacks and appreciating our work. Following includes our replies and explanations regarding the changes in the revised draft based on the reviewer's feedback:
>
> **“C1, diagram inconsistency”**: We have inspected the subfigures of Fig. 5 to make sure that they are aligned with Eq. (1) and (2) and have made following changes:
> - In Fig.5(b) (Text as CoT), we have included an arrow from the text output $\mathbf{O_{text}}$ back to the model as an input to show that $\mathbf{O_{traj}}$ is also conditioned on $\mathbf{O_{text}}$. We have done similar for Fig.5(c) (Trajectory as CoT). Please note that, for consistency across the paper, we have employed the same arrow design which we use to represent the autoregressive generation in Fig. 6(c).
> - We have noted an inconsistency/typo in Fig.5(c)(Trajectory as CoT), where $\mathbf{O_{inittraj}}$ was conditioned on $\mathbf{O_{text}}$ instead of the observations $\mathbf{X}$ in the initial draft. In this version, we have also fixed this typo.
> - We have also extended the caption of the Fig.5 for making it clearer.
> - Finally, we have made some cosmetic changes in the figure: (i) We have increased the height of the purple “Foundation Model” boxes, where the output arrows start after the last input such that the height provides a sense of time. (ii) We have included navigation icons (colored circles) for the subgroups.
>
> **“C2, about paper selection protocol”:** We considered papers published in peer-reviewed venues until we submitted our paper (October 2025). Additionally, we include a few representative preprints to provide a broader discussion. This selection protocol was stated in the Footnote 2 in the first submission of the draft.  To make it more visible, we have removed Footnote 2 and included a more detailed paragraph as the last paragraph of Sec.1.1 (after contributions).
>
> **“C3, missing discussion in taxonomy groups”:** We have included “Summary of tradeoffs" boxes as per-leaf-node summary at the end each subsection where we discuss the approaches in Sections 4 and 5. In these boxes, we assess each of the subgroups from the perspectives of (i) data needs (including the reliance on language supervision), (ii) inference efficiency (also from the perspective of deployment feasibility), (iii) explainability and (iv) capabilities.
>
> **“C4, missing citations”:** We have added citations and references in the text for all datasets, methods and benchmarks previously mentioned only in a table. This includes references to DAIR-V2X, BART, UL2, Flan-T5, NAVSIM v2, LAV, CARLA Longest6, and CARLA Town05. Additionally, we also replaced the citation of two preprints (DriveMLM and OpenDriveVLA) as they have been accepted to peer-reviewed venues since our initial submission.
>
> **“M1, taxonomy presentation too abstract”:** We have added representative method examples while we discuss each leaf node in Sections 3.1 and 3.2.
>
> **“M2, data curation with concrete dataset examples”:**  In data curation paragraph of Sec 4.1, we have used bullet points that align with four different types of datasets we represent in Fig 6(a). For each type of dataset, we have included and cited a characteristic dataset.
>
> **“M3, missing depth in asset availability”:** We have updated Sec. 6 with a high-level synthesis of asset availability. For this, we have included Fig. 9 (an improved version of Fig.A.11 in Appendix of the initial submission) in Sec. 6 to demonstrate the openness distribution of each type of asset in a clearer and summarised form. Based on this figure, we now discuss the reproducibility gap caused by missing training code and model weights. Furthermore, we add a high-level summary of data availability, highlighting synthetic versus real-world data as a characteristic. If you were expecting different analyses, we would appreciate it if you could kindly let us know explicitly, and we will do our best to address your concern.

---

### Review · Reviewer_mqh2 · 2026-01-16

**Summary Of Contributions:**

This survey provides a structured overview of how foundation models are being applied to autonomous driving trajectory planning.

1. It proposes a clear taxonomy that separates approaches where foundation models are tailored for trajectory planning versus those where foundation models guide planning, with finer subcategories (e.g., language/trajectory chain-of-thought, interaction, distillation);
2. It compiles and reviews a broad set of recent methods with comparative tables/figures;
3. It offers practical guidance on adapting foundation models to planning, including data curation, trajectory representations, and fine-tuning strategies;
4. It introduces an openness/reproducibility assessment framework (with explicit scoring levels) to characterize the availability of code/data/weights and highlight reproducibility gaps, alongside a discussion of open challenges for deployable FM-based planners.

**Audience:**

Yes

**Audience Explanation:**

Many readers would likely find this survey interesting because it makes a fast-moving and fragmented area much easier to navigate.

It offers a clear taxonomy of how foundation models are used for autonomous driving trajectory planning, backs it with a curated and structured coverage of recent methods (with tables/figures), and includes practical guidance on adaptation choices like data curation, trajectory representations, and fine-tuning.

It’s also especially useful for researchers and practitioners who care about reproducibility and deployment, since it proposes an explicit openness scoring framework and discusses real-world constraints such as inference cost/latency (including CoT overhead) versus real-time needs.

**Claims And Evidence:**

Yes

**Claims Explanation:**

I find the survey’s key claims convincing:

1. The authors lay out an explicit taxonomy (e.g., foundation models tailored for trajectory planning vs guiding planning) and then support it with a broad, organized coverage of recent work summarized through structured discussion and tables/figures.

2. Their reproducibility/openness conclusions are particularly solid because they use a clearly defined scoring rubric (with concrete levels and guidance for edge cases such as licensing), so the assessment is transparent and checkable rather than anecdotal.

3. They also strike the right tone in how they interpret results: the survey explicitly flags confounders that make cross-paper performance comparisons tricky, which reduces the risk of over-claiming and makes the takeaways feel more trustworthy.

**Requested Changes:**

While the survey is already strong, I would encourage the authors to (i) explicitly describe the literature collection/inclusion methodology, and (ii) add lightweight comparability and taxonomy-assignment annotations in the main tables (e.g., evaluation setting, dataset/metric flags, and brief rationale for categorization), which would further improve transparency and prevent readers from over-interpreting cross-paper comparisons.

---

> ### Author Response · Authors · 2026-01-29
> **Replies to Reviewer mqh2**
>
> Thanks for appreciating our work and constructive feedback. Following includes our replies and explanations regarding the changes in the revised draft based on the reviewer's feedback:
>
> **About paper selection protocol**: We considered papers published in peer-reviewed venues until we submitted our paper (October 2025). Additionally, we include a few representative preprints to provide a broader discussion. This selection protocol was stated in the Footnote 2 in the first submission of the draft.  To make it more visible, we have removed Footnote 2 and included a more detailed paragraph as the last paragraph of Section 1.1 (after contributions).
>
> **Lightweight comparability and taxonomy-assignment annotations in the main tables**: For improved clarity and navigation of taxonomy-assignment annotations in main tables, we have added taxonomy icons (as colored circles) to the methods in Table 1 and Table 6.

---

### Author Response · Authors · 2026-01-29
**General Comment**

We thank all the reviewers for their constructive comments and for appreciating our contributions.

All reviewers commended our hierarchical taxonomy and openness/reproducibility assessment. Specifically, Reviewer mqh2 mentioned that *“They also strike the right tone in how they interpret results”*, t9if praised our taxonomy to be *“well-structured taxonomy that brings order to a fragmented literature”*, and t9if commended the useful discussion of practical design trade-offs.

**We have submitted a revision of our paper.** In the submitted revised version, we have used blue font for the additions and red strikethrough font for the removals to make it easy to see the difference.

In summary, considering the reviewers' comments;

- **We have made Fig. 5, where we present the formulations of the taxonomy, precise** by showing the autoregressive decoding in (b) and (c) explicitly.
- To make the **paper selection protocol** more explicit, we have removed Footnote 2 and included a paragraph in Section 1.1 where we discuss the paper selection protocol in detail. To address the concern of Reviewer t9if, we also clarified in that paragraph that we don’t aim to compare the methods quantitatively but conceptually compare them.
- We have included **“Summary of trade-offs" boxes** as per-leaf-node summary at the end each subsection where we discuss the approaches in Sec.4 and Sec.5.
- We have included and cited **example methods** in Sec. 3, and datasets in Sec. 4.1 data curation, to avoid being too abstract.
- We have carefully checked the datasets and models mentioned in our paper (including all tables), and **completed missing citations**.
- To provide a clearer high-level synthesis of available assets in the openness discussion (Sec.6), we have included Fig.9, and based on this figure, we have provided a **more discussion on the availabilities of different type of assets.**
- Given that we already discuss some emerging directions such as Agentic AI, we have renamed Sec.7 “Open Issues” as **“Open Issues and Emerging Directions”** and extended our discussions in Section 7 to include more emerging directions such as the potential benefits of the world models.
- We have **clarified the boundaries** between CoT-based planning vs. language-interaction capabilities in Sec.3 also by providing examples.

In the following, we provide detailed explanations for these revisions while addressing the individual concerns of each reviewer, alleviating any misconceptions there maybe.

---

### Decision · Action_Editor_iQMD · 2026-03-01

**Recommendation:** Accept as is

**Additional Comments:**

This manuscript surveys multi-modal foundation models (FMs) for autonomous driving with a focus on trajectory planning, where models infer motion trajectories from sensory inputs, including language-conditioned variants such as Vision-Language-Action (VLA). It distinguishes FMs that directly predict trajectories from those that guide trajectory planning. The authors present a unifying taxonomy and systematically evaluate 37 recent approaches, comparing architectures, methodological strengths, capabilities and limitations, and open-access availability.

---

Three reviews have been collected. All reviewers evaluate this manuscript positively. Key contributions and **strengths** of this work include:

- (S1) Well-structured and useful taxonomy with basic formalization. The taxonomy and survey make a fast-moving and fragmented area easier to navigate.
- (S2) Broad, organized coverage of recent work.
- (S3) Providing practical guidance, including detailed technical overview, practical design trade-offs, and assessment on reproducibility.
- (S4) Useful discussion and assessment  of reproducibility and openness.
- (S5) Convincing interpretation of results and discussion of open challenges.


---

**Review process:** The reviewers provided valuable comments, based on which the authors could revise and improve their manuscript. The reviewers' concerns were addressed. I thank both the authors and reviewers for their active engagement throughout this process.

**Summary of evaluation:** Based on the reviews and the outcome of the discussion phase, the manuscript is suitable and ready for publication. Based on the above mentioned strengths, the manuscript is a candidate for "Survey Certification".

**Audience:**

Yes

**Audience Explanation:**

The topic of this survey is of interest to multiple readers, including those interested in foundation models, autonomous driving, multimodal learning, and decision-making systems. Reviewers praise its usefulness as a practical guide for navigating a fast-developing and fragmented area.

**Claims And Evidence:**

Yes

**Claims Explanation:**

The manuscript surveys recent work on foundation models in autonomous driving. Its claims concern taxonomy, categorization, and synthesis of existing work rather than new empirical results. All reviewers agree these claims are well supported.